# Learning the Dynamics of Physical Systems from Sparse Observations with Finite Element Networks

**Marten Lienen & Stephan Günnemann**
Department of Informatics & Munich Data Science Institute
Technical University of Munich, Germany
{marten.lienen,guennemann}@in.tum.de

## Abstract

We propose a new method for spatio-temporal forecasting on arbitrarily distributed points. Assuming that the observed system follows an unknown partial differential equation, we derive a continuous-time model for the dynamics of the data via the finite element method. The resulting graph neural network estimates the instantaneous effects of the unknown dynamics on each cell in a meshing of the spatial domain. Our model can incorporate prior knowledge via assumptions on the form of the unknown PDE, which induce a structural bias towards learning specific processes. Through this mechanism, we derive a transport variant of our model from the convection equation and show that it improves the transfer performance to higher-resolution meshes on sea surface temperature and gas flow forecasting against baseline models representing a selection of spatio-temporal forecasting methods. A qualitative analysis shows that our model disentangles the data dynamics into their constituent parts, which makes it uniquely interpretable.

## 1 Introduction

The laws driving the physical world are often best described by partial differential equations (PDEs) that relate how a magnitude of interest changes in time with its change in space. They describe how the atmosphere and oceans circulate and interact, how structures deform under load and how electromagnetic waves propagate (Courant & Hilbert, 2008). Knowledge of these equations lets us predict the weather (Coiffier, 2011), build sturdier structures, and communicate wirelessly. Yet, in many cases we only know the PDEs governing a system partially (Isakov, 2006) or not at all, or solving them is too computationally costly to be practical (Ames, 2014).

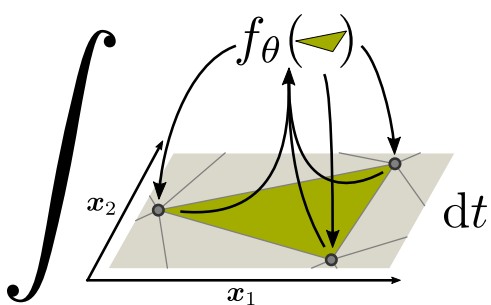

Figure 1: Finite Element Networks predict the instantaneous change of each node by estimating the effect of the unknown generating dynamics on the domain volume that the node shares with its neighbors.

Machine learning researchers try to fill in these gaps with models trained on collected data. For example, neural networks have been trained for weather forecasts (Shi et al., 2015) and fluid flow simulations (Belbute-Peres et al., 2020), both of which are traditionally outcomes of PDE solvers. Even the dynamics of discrete dynamical systems such as traffic (Li et al., 2018) and crowds (Zhang et al., 2017) have been learned from data. A challenge facing these models is the high cost of acquiring training data, so the data is usually only available sparsely distributed in space. Since graphs are a natural way to structure sparse data, models incorporating graph neural networks (GNNs) have been particularly successful for spatio-temporal forecasting (Yu et al., 2018; Wu et al., 2019).

Our implementation is available at https://www.daml.in.tum.de/finite-element-networks/

In the domain of physical processes we can reasonably assume that the observed system follows a PDE. There are mainly two ways to incorporate this assumption as a-priori knowledge into a model. First, we can encode a known PDE into a loss function that encourages the model to fulfill the equation (Raissi et al., 2019). Another way to go about this is to derive the model structure itself from known laws such as the convection-diffusion equation (de Bézenac et al., 2018). We will follow the second approach.

Consider a dynamical system on a bounded domain $\Omega \subset \mathbb{R}^d$ that is governed by the PDE

$$\partial_t u = F\left(t, \boldsymbol{x}, u, \partial_{\boldsymbol{x}} u, \partial_{\boldsymbol{x}}^2 u, ...\right) \tag{1}$$

on functions $u : [0, T] \times \Omega \to \mathbb{R}^m$. If we have a dense measurement $u_0 : \Omega \to \mathbb{R}^m$ of the current state of the system and a solution $u$ that satisfies Eq. (1) for all $t \in [0, T]$ and also fulfills the initial condition $u(0, \boldsymbol{x}) = u_0(\boldsymbol{x})$ at all points $\boldsymbol{x} \in \Omega$, we can use $u$ as a forecast for the state of the system until time $T$. From a spatio-temporal forecasting perspective, this means that we can forecast the evolution of the system if we have a continuous measurement of the state, know the dynamics $F$, and can find solutions of Eq. (1) efficiently. Unfortunately, in practice we only have a finite number of measurements at arbitrary points and only know the dynamics partially or not at all.

**Contributions.** An established numerical method for forecasts in systems with fully specified dynamics is the finite element method (FEM) (Brenner et al., 2008). In this paper, we introduce the first graph-based model for spatio-temporal forecasting that is derived from FEM in a principled way. Our derivation establishes a direct connection between the form of the unknown dynamics and the structure of the model. Through this connection our model can incorporate prior knowledge on the governing physical processes via assumptions on the form of the underlying dynamics. We employ this mechanism to derive a specialized model for transport problems from the convection equation. The way that the model structure arises from the underlying equation makes our models uniquely interpretable. We show that our transport model disentangles convection and the remainder of the learned dynamics such as source/sink behavior, and that the activations of the model correspond to a learned flow field, which can be visualized and analyzed. In experiments on multi-step forecasting of sea surface temperature and gas flow, our models are competitive against baselines from recurrent, temporal-convolutional, and continuous-time model classes and the transport variant improves upon them in the transfer to higher-resolution meshes.

## 2 BACKGROUND

### 2.1 FINITE ELEMENT METHOD

In the following, we will outline how to approximate a solution $u$ to the dynamics in Eq. (1) from an initial value $u_0$ by discretizing $u$ in space using finite elements. Let $\mathcal{X}$ be a set of points with a triangulation $\mathcal{T}$ of $d$-dimensional, non-overlapping simplices

$$\mathcal{X} = \{\boldsymbol{x}^{(i)} \in \mathbb{R}^d\}_{i=1}^N \qquad \mathcal{T} = \{\Delta^{(j)} \mid \Delta^{(j)} \subset \mathcal{X}, |\Delta^{(j)}| = d+1\}_{j=1}^{N_{\mathcal{T}}} \tag{2}$$

such that $\cup_{\Delta \in \mathcal{T}} \mathrm{CH}(\Delta)$ equals the domain $\Omega$ where $\mathrm{CH}(\Delta)$ is the convex hull of simplex $\Delta$. So we define a simplex $\Delta^{(j)} \in \mathcal{T}$ representing the $j$-th mesh cell as the set of vertices of the cell and denote the domain volume covered by the cell by the convex hull $\mathrm{CH}(\Delta^{(j)})$ of the vertices. We will assume $u$ to be a scalar field, i.e. $u : [0, T] \times \Omega \to \mathbb{R}$. If $u$ is a vector field, we treat it as a system of $m$ scalar fields instead. For a detailed introduction to FEM, we refer the reader to Igel (2017).

**Basis Functions.** A priori, we assume that the unknown solution $u$ to our problem lies in an infinite-dimensional function space $\mathcal{U}$. The first step in FEM to make the problem numerically feasible is to approximate $\mathcal{U}$ with a finite-dimensional linear subspace $\tilde{\mathcal{U}}$. This subspace can then be written in terms of linear combinations of basis functions $\tilde{\mathcal{U}} = \mathrm{span}\left\{\varphi^{(1)}, ..., \varphi^{(N)}\right\}$. There are many possible bases and the choice determines various qualities of the resulting procedure such as continuity of the approximation and the sparsity pattern of the mass matrix in Eq. (7).

In our case, we choose the so-called P1 basis of piecewise linear functions (hat functions), see Fig. 2a (Igel, 2017). There are as many basis functions as there are points and each is uniquely defined by being linear when restricted to a single cell $\Delta \in \mathcal{T}$ and the constraint

$$\varphi^{(j)}(\boldsymbol{x}^{(i)}) = \begin{cases} 1 & \text{if } \boldsymbol{x}^{(i)} = \boldsymbol{x}^{(j)} \\ 0 & \text{otherwise} \end{cases} \qquad \forall \boldsymbol{x}^{(i)} \in \mathcal{X}. \tag{3}$$

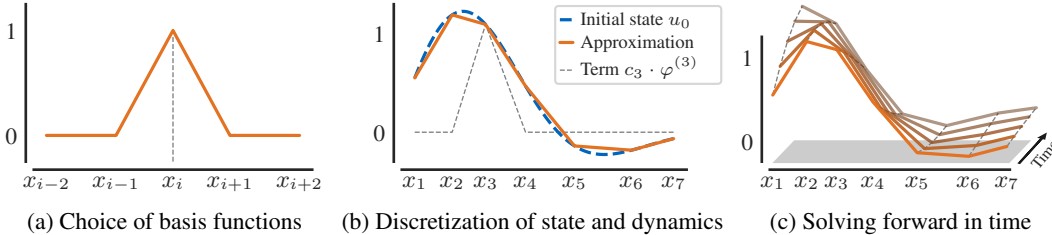

(a) Choice of basis functions     (b) Discretization of state and dynamics     (c) Solving forward in time

Figure 2: Solving a PDE with the Galerkin method and method of lines consists of three steps.

So the basis function $\varphi^{(j)}$ is 1 at $\boldsymbol{x}^{(j)}$, falls linearly to 0 on mesh cells adjacent to $\boldsymbol{x}^{(j)}$ and is 0 everywhere else. The resulting finite-dimensional function space $\tilde{\mathcal{U}}$ is the space of linear interpolators between values at the vertices, see Fig. 2b. An important property is that if we expand $u \in \tilde{\mathcal{U}}$ in this basis, the value of $u$ at the $i$-th node is just its $i$-th coefficient.

$$u(\boldsymbol{x}^{(i)}) = \sum_{j=1}^{N} c_j \varphi^{(j)}(\boldsymbol{x}^{(i)}) = c_i \tag{4}$$

**Galerkin Method.** A piecewise linear approximation $u \in \tilde{\mathcal{U}}$ is not differentiable everywhere and therefore cannot fulfill Eq. (1) exactly. So instead of requiring an exact solution, we ask that the residual $R(u) = \partial_t u - F(t, \boldsymbol{x}, u, ...)$ be orthogonal to the approximation space $\tilde{\mathcal{U}}$ with respect to the inner product $\langle u, v \rangle_\Omega = \int_\Omega u(\boldsymbol{x}) \cdot v(\boldsymbol{x}) \, \mathrm{d}\boldsymbol{x}$ at any fixed time $t$. In effect we are looking for the best possible solution in $\tilde{\mathcal{U}}$. Because $\tilde{\mathcal{U}}$ is generated by a finite basis, the orthogonality requirement decomposes into $N$ equations, one for each basis function.

$$\langle R(u), v \rangle_\Omega = 0 \quad \forall v \in \tilde{\mathcal{U}} \qquad \Longleftrightarrow \qquad \langle R(u), \varphi^{(i)} \rangle_\Omega = 0 \quad \forall i = 1, ..., N \tag{5}$$

Plugging the residual back in and using the linearity of the inner product, we can reconstruct a system of equations that resemble the PDE that we started with.

$$\langle \partial_t u, \varphi^{(i)} \rangle = \langle F(t, \boldsymbol{x}, u, ...), \varphi^{(i)} \rangle_\Omega \quad \forall i = 1, ..., N \tag{6}$$

At this point we can stack the system of $N$ equations into a vector equation. If we plug in the basis expansion $\sum_{j=1}^{N} c_j \varphi^{(j)}$ for $u$ into the left hand side, we get a linear system

$$\boldsymbol{A} \partial_t \boldsymbol{c} = \boldsymbol{m} \tag{7}$$

where $A_{ij} = \langle \varphi^{(i)}, \varphi^{(j)} \rangle_\Omega$ is the so called mass matrix, $\boldsymbol{c}$ is the vector of basis coefficients of $u$, and $m_i = \langle F(t, \boldsymbol{x}, u, ...), \varphi^{(i)} \rangle_\Omega$ captures the effect of the dynamics $F$. The left hand side evaluates to $\boldsymbol{A} \partial_t \boldsymbol{c}$, because the basis functions are constant with respect to time. The right hand side cannot be further simplified without additional assumptions on $F$.

**Method of Lines.** If we can evaluate the right hand side $\boldsymbol{m}$, we can solve the linear system in Eq. (7) for the temporal derivatives of the coefficients of $u$ at each point in time. In fact we have converted the PDE into a system of ordinary differential equations (ODEs) which we can solve with an arbitrary ODE solver given an initial value $\boldsymbol{c}^{(0)}$ as in Fig. 2c. This is known as the method of lines because we solve for $u$ along parallel lines in time.

To find a vector field $u : [0, T] \times \Omega \to \mathbb{R}^m$ instead of a scalar field, we treat the $m$ dimensions of $u$ as a system of $m$ scalar fields. This results in $m$ copies of Eq. (7), which we need to solve simultaneously. Because the mass matrix $\boldsymbol{A}$ is constant with respect to $u$, we can combine the system into a matrix equation

$$\boldsymbol{A} \partial_t \boldsymbol{C} = \boldsymbol{M} \tag{8}$$

where $\boldsymbol{C}, \boldsymbol{M} \in \mathbb{R}^{N \times m}$ are the stacked $\boldsymbol{c}$ and $\boldsymbol{m}$ vectors, respectively. In summary, the spatial discretization with finite elements allows us to turn the PDE (1) into the matrix ODE (8).

## 2.2 MESSAGE PASSING NEURAL NETWORKS

Message-Passing Neural Networks (MPNNs) are a general framework for learning on graphs that encompass many variants of graph neural networks (Gilmer et al., 2017). It prescribes that nodes in

a graph iteratively exchange messages and update their state based on the received messages for $P$ steps. For a graph $\mathcal{G} = (\mathcal{V}, \mathcal{E})$ with nodes $\mathcal{V}$ and edges $\mathcal{E}$, and initial node states $\boldsymbol{h}_v^{(0)} \; \forall v \in \mathcal{V}$, the $p$-th propagation step is

$$\boldsymbol{h}_v^{(p)} = f_{\text{upd}} \left( \boldsymbol{h}_v^{(p-1)}, \sum_{\{u,v\} \in \mathcal{E}} f_{\text{msg}} \left( \boldsymbol{h}_u^{(p-1)}, \boldsymbol{h}_v^{(p-1)} \right) \right), \tag{9}$$

where $f_{\text{msg}}$ maps node states and edge attributes to messages and $f_{\text{upd}}$ updates a node's state with the aggregated incoming messages. The final node states $\boldsymbol{h}_v^{(P)}$ can then be interpreted as per-node predictions directly or passed as node embeddings to downstream systems.

In this work, we employ a slight generalization of the above to undirected hypergraphs, i.e. graphs where the edges are sets of an arbitrary number of nodes instead of having a cardinality of exactly 2. For such a hypergraph $\mathcal{G} = (\mathcal{V}, \mathcal{E})$ with nodes $\mathcal{V}$ and hyperedges $\varepsilon = \{u, v, w, ...\} \in \mathcal{E}$, and initial node states $\boldsymbol{h}_v^{(0)} \; \forall v \in \mathcal{V}$, the $p$-th propagation step is

$$\boldsymbol{h}_v^{(p)} = f_{\text{upd}} \left( \boldsymbol{h}_v^{(p-1)}, \sum_{\substack{\varepsilon \in \mathcal{E} \\ \text{s.t.} v \in \varepsilon}} f_{\text{msg}} \left( \left\{ \boldsymbol{h}_u^{(p-1)} \mid u \in \varepsilon \right\} \right)_v \right). \tag{10}$$

Note that $f_{\text{msg}}$ jointly computes a separate message for each node $v$ participating in a hyperedge $\varepsilon$.

## 3 FINITE ELEMENT NETWORKS

Consider a set of nodes $\mathcal{X} = \{\boldsymbol{x}^{(i)} \in \mathbb{R}^d\}_{i=1}^N$ representing $N$ points in $d$-dimensional space. At each node we measure $m$ features $\boldsymbol{y}^{(t_0,i)} \in \mathbb{R}^m$. Our goal is to predict $\boldsymbol{y}^{(t_j)}$ at future timesteps $t_j$ by solving PDE (1). For it to be well-defined, we need a continuous domain encompassing the nodes $\mathcal{X}$.

By default, we construct such a domain by Delaunay triangulation. The Delaunay algorithm triangulates the convex hull of the nodes into a set of mesh cells – $d$-dimensional simplices – $\mathcal{T} = \{\Delta^{(i)} \subset \mathcal{X} \mid |\Delta^{(i)}| = d + 1\}_{i=1}^{N_\mathcal{T}}$ which we represent as sets of nodes denoting the cell's vertices. The PDE will then be defined on the domain $\Omega = \cup_{\Delta \in \mathcal{T}} \text{CH}(\Delta)$. One shortcoming of this algorithm is that it can produce highly acute, sliver-like cells on the boundaries of the domain if interior nodes are very close to the boundary of the convex hull. We remove these slivers in a post-processing step, because they have small area and thus contribute negligible amounts to the domain volume, but can connect nodes which are far apart. See Appendix H for our algorithm. As an alternative to Delaunay triangulation, a mesh can be explicitly specified to convey complex geometries such as disconnected domains, holes in the domain and non-convex shapes in general.

As a first step towards solving the PDE, we need to represent $\boldsymbol{y}^{(t_0)}$ as a function on $\Omega$. We define the function $u_0(\boldsymbol{x})_k = \sum_{i=1}^N y_k^{(t_0)} \varphi^{(i)}(\boldsymbol{x})$, such that its coefficients as a vector in the P1 basis encode the data and evaluating it at the nodes $u_0(\boldsymbol{x}^{(i)}) = \boldsymbol{y}^{(t_0,i)}$ agrees with the data by construction.

In Section 2.1, we have seen that solving PDE (1) approximately under an initial condition $u(t_0, \boldsymbol{x}) = u_0(\boldsymbol{x})$ can be cast as an initial value ODE problem over the trajectory of the coefficients of a solution in time. Above, we encoded the features $\boldsymbol{y}^{(t_0)}$ in the coefficients of $u_0$, meaning that Eq. (8) exactly describes the trajectory of the features through time. The dynamics of the features are thus given by

$$\boldsymbol{A} \partial_t \boldsymbol{Y}^{(t)} = \boldsymbol{M} \tag{11}$$

where $Y_{ik}^{(t)} = y_k^{(t,i)}$ is the feature matrix and $A_{ij} = \langle \varphi^{(i)}, \varphi^{(j)} \rangle_\Omega$ is the mass matrix. We call the matrix $\boldsymbol{M}$ on the right hand side the message matrix and it is given by

$$M_{ik} = \langle F(t, \boldsymbol{x}, u, ...)_k, \varphi^{(i)} \rangle_\Omega \tag{12}$$

where $u$ encodes the predicted features $\boldsymbol{Y}$ at time $t$ in its coefficients.

Our goal is to solve the feature dynamics (11) with an ODE solver forward in time to compute the trajectory of $\boldsymbol{Y}$ and thereby predict $\boldsymbol{y}^{(t_j)}$. To apply an ODE solver, we have to compute $\partial_t \boldsymbol{Y}$ at each time $t$ which in turn consists of, first, evaluating the right hand side and, second, solving the resulting sparse linear system.

Solving Eq. (11) for $\partial_t \boldsymbol{Y}$ with the exact sparse mass matrix $\boldsymbol{A}$ carries a performance penalty on GPUs, because sparse matrix solving is difficult to parallelize. To avoid this operation, we lump the mass matrix. Lumping of the mass matrix is a standard approximation in FEM for time-dependent PDEs to reduce computational effort (Lapidus & Pinder, 1999), which leads to good performance in practice.

$$\tilde{\boldsymbol{A}}_{ii} = \sum\nolimits_{j=1}^{N} \langle \varphi^{(i)}, \varphi^{(j)} \rangle_{\Omega} \tag{13}$$

We employ the direct variant that diagonalizes the mass matrix by row-wise summation, making the inversion of $\boldsymbol{A}$ trivial.

The remaining step in solving Eq. (11) is evaluating the right hand side. Let $\mathcal{T}_i = \left\{ \Delta \in \mathcal{T} \mid \boldsymbol{x}^{(i)} \in \Delta \right\}$ be the set of all mesh cells that are adjacent to node $\boldsymbol{x}^{(i)}$. Then we can break the elements of $\boldsymbol{M}$ up into the contributions from individual cells adjacent to each node.

$$M_{ik} = \langle F\left(t, \boldsymbol{x}, u, ...\right)_k, \varphi^{(i)} \rangle_{\Omega} = \sum\nolimits_{\Delta \in \mathcal{T}_i} \langle F\left(t, \boldsymbol{x}, u, ...\right)_k, \varphi^{(i)} \rangle_{\mathrm{CH}(\Delta)}. \tag{14}$$

The sum terms capture the effect that the dynamics have on the state of the system such as convection and chemical reaction processes given the geometry of the mesh cell, the values at the nodes, and the basis function to integrate against. However, the dynamics $F$ are of course unknown and need to be learned from data.

Yet, if we were to introduce a deep model $f_\theta \approx F$ in Eq. (14), we would run into two problems. First, each evaluation would require the numerical integration of a neural network over each mesh cell against multiple basis functions and adaptive-step ODE solvers can require hundreds of evaluations. Second, conditioning $f_\theta$ on just $t$, $\boldsymbol{x}$ and $u(\boldsymbol{x})$ would not provide the model with any spatial information. Such a model would not be able to estimate spatial derivatives $\partial_{\boldsymbol{x}} u$ internally and could therefore only represent PDEs that are actually just ODEs.

We deal with both problems at once by factoring the inner products in Eq. (14) into

$$\langle F\left(t, \boldsymbol{x}, u, ...\right)_k, \varphi^{(i)} \rangle_{\mathrm{CH}(\Delta)} = F_{\Delta,k}^{(i)} \cdot \langle 1, \varphi^{(i)} \rangle_{\mathrm{CH}(\Delta)}. \tag{15}$$

Such a scalar coefficient $F_{\Delta,k}^{(i)}$ always exists, because $\langle 1, \varphi^{(i)} \rangle_{\mathrm{CH}(\Delta)}$ is constant with respect to $t$ and $u$ and never $0$. On the basis of Eq. (15), we can now introduce a deep model $f_\theta \approx F_{\Delta}^{(i)}$. With this, we sidestep the numerical integration of a neural network, because $F_{\Delta}^{(i)}$ is constant per mesh cell and the coefficients $\langle 1, \varphi^{(i)} \rangle_{\mathrm{CH}(\Delta)}$ can be precomputed once per domain. At the same time, $F_{\Delta}^{(i)}$ depends jointly on all $\boldsymbol{x}$ in a mesh cell as well as their solution values $u(\boldsymbol{x})$. Therefore our deep model $f_\theta$ naturally also needs to be conditioned on these spatially distributed values, giving it access to spatial information and making it possible to learn actual PDE dynamics.

### 3.1 MODEL

With these theoretical foundations in place, we can formally define **Finite Element Networks (FENs)**. Let $\mathcal{G} = (\mathcal{X}, \mathcal{T})$ be the undirected hypergraph with nodes $\mathcal{X}$ and hyperedges $\mathcal{T}$ corresponding to the meshing of the domain, so each mesh cell $\Delta \in \mathcal{T}$ forms a hyperedge between its vertices. Let $f_\theta$ be a neural network that estimates the cell-wise dynamics $F_\Delta$ from time, cell location, cell shape and the values at the vertices

$$f_{\theta,\Delta}^{(t,i)} := f_\theta \left( t, \boldsymbol{\mu}_\Delta, \boldsymbol{x}_\Delta, \boldsymbol{y}_\Delta^{(t)} \right)^{(i)} \approx F_\Delta^{(i)} \tag{16}$$

where $\boldsymbol{\mu}_\Delta$ is the center of cell $\Delta$, $\boldsymbol{x}_\Delta = \{ \boldsymbol{x}^{(i)} - \boldsymbol{\mu}_\Delta \mid \boldsymbol{x}^{(i)} \in \Delta \}$ are the local coordinates of the cell vertices and $\boldsymbol{y}_\Delta^{(t)} = \{ \boldsymbol{y}^{(t,i)} \mid \boldsymbol{x}^{(i)} \in \Delta \}$ are the features at the vertices at time $t$. We call $f_\theta$ the free-form term, because it does not make any assumptions on the form of $F$ and can in principle model anything such as reactions between features or sources and sinks. $f_\theta$ conditions on the cell centers $\boldsymbol{\mu}_\Delta$ and the local vertex coordinates $\boldsymbol{x}_\Delta$ separately to help the model distinguish between cell position and shape, and improve spatial generalization.

Define the following message and update functions in the MPNN framework

$$f_{\mathrm{msg}}\left(\Delta\right)_{\boldsymbol{x}^{(i)}} = f_{\theta,\Delta}^{(t,i)} \cdot \langle 1, \varphi^{(i)} \rangle_{\mathrm{CH}(\Delta)} \qquad f_{\mathrm{upd}}\left(\boldsymbol{x}^{(i)}, \boldsymbol{m}^{(i)}\right) = \tilde{\boldsymbol{A}}_{ii}^{-1} \boldsymbol{m}^{(i)} \tag{17}$$

where $\boldsymbol{m}^{(i)} = \sum_{\Delta \in \mathcal{T}} f_{\text{msg}}(\Delta)_{\boldsymbol{x}^{(i)}}$ are the aggregated messages for node $\boldsymbol{x}^{(i)}$. Performing one message passing step ($P = 1$) in this model is exactly equivalent to solving for the feature derivatives $\partial_t \boldsymbol{Y}^{(t)}$ in Eq. (11) with the lumped mass matrix from Eq. (13) and the message matrix in Eq. (14) with the factorization from Eq. (15) and $f_\theta$ as a model for $F_\Delta$. So making a forecast with FEN for $T$ points in time $\boldsymbol{t} = (t_0, t_1, ..., t_T)$ based on a sparse measurement $\boldsymbol{y}^{(t_0)}$ means solving an ODE where at each evaluation step $t_e$ we run one message passing step in the MPNN described above to compute the current time derivatives of the features $\partial_t \boldsymbol{y}|_{t=t_e}$.

$$\hat{\boldsymbol{y}}^{(t_0, t_1, ..., t_T)} = \text{ODESolve}(\boldsymbol{y}^{(t_0)}, \partial_t \boldsymbol{y}, \boldsymbol{t}) \tag{18}$$

Because FENs model the continuous-time dynamics of the data, they can also be trained on and make predictions at irregular timesteps.

By controlling the information available to $f_\theta$, we can equip the model with inductive biases. If we omit the current time $t$, the learned dynamics become necessarily autonomous. Similarly, if we do not pass the cell position $\boldsymbol{\mu}_\Delta$, the model will learn stationary dynamics.

An orthogonal and at least as powerful mechanism to induce inductive biases into the model is through assumptions on the form of the unknown dynamics $F$. Let us assume that the dynamics are not completely unknown, but that a domain expert told us that the generating dynamics of the data include a convective component. So the dynamics for feature $k$ are of the form

$$F(t, \boldsymbol{x}, u, ...)_k = -\nabla \cdot \left( v^{(k)}(t, \boldsymbol{x}, u, ...)u_k \right) + F'(u, \partial_{\boldsymbol{x}} u, ...)_k \tag{19}$$

where $v^{(k)}(t, \boldsymbol{x}, u, ...) \in \mathbb{R}^d$ is the divergence-free velocity field of that feature and $F'$ represents the still unknown remainder of the dynamics. We assume the velocity field to be divergence-free, because we want the velocity field to model only convection and absorb sources and sinks in the free-form term.

By following similar steps with the convection term $-\nabla \cdot \left( v^{(k)}(t, \boldsymbol{x}, u, ...)u_k \right)$ as for the unknown dynamics, we arrive at a specialized term for modeling convection. Let $g_\theta$ be a neural network that estimates one velocity vector per cell and attribute.

$$g_{\theta, \Delta}^{(t,i)} := g_\theta \left( t, \boldsymbol{\mu}_\Delta, \boldsymbol{x}_\Delta, \boldsymbol{y}_\Delta^{(t)} \right) \approx v(t, \boldsymbol{x}, u, ...) \in \mathbb{R}^{m \times d}. \tag{20}$$

Ensuring that the learned velocity field is globally divergence-free would be a costly operation, but by sharing the same predicted velocity vector between all vertices of each cell, we can cheaply guarantee that the velocity field is at least locally divergence-free within each cell. The corresponding message function, which we derive in Appendix B, is

$$f_{\text{msg}}^v(\Delta)_{\boldsymbol{x}^{(i)}} = \sum_{\boldsymbol{x}^{(j)} \in \Delta} \boldsymbol{y}^{(t,j)} \odot \left( g_{\theta, \Delta}^{(t,i)} \cdot \langle \nabla \varphi^{(j)}, \varphi^{(i)} \rangle_{\text{CH}(\Delta)} \right). \tag{21}$$

With this we can extend FEN with specialized transport-modeling capabilities by adding the two message functions and define **Transport-FEN (T-FEN)** as a FEN with the message function

$$f_{\text{msg}}^{\text{TFEN}}(\Delta)_{\boldsymbol{x}^{(i)}} = f_{\text{msg}}^v(\Delta)_{\boldsymbol{x}^{(i)}} + f_{\text{msg}}(\Delta)_{\boldsymbol{x}^{(i)}}. \tag{22}$$

Consequently, T-FEN learns both a velocity field to capture convection and a free-form term that fits the unknown remainder of the dynamics.

**Network Architecture.** We instantiate both $f_\theta$ and $g_\theta$ as multilayer perceptrons (MLPs) with $\tanh$ non-linearities. The input arguments are concatenated after sorting the cell vertices by the angle of $\boldsymbol{x}^{(i)} - \boldsymbol{\mu}_\Delta$ in polar coordinates. This improves generalization because $f_\theta$ and $g_\theta$ do not have to learn order invariance of the nodes. Both MLPs have an input dimension of $1 + d + (d + 1) \cdot (m + d)$ in the non-autonomous, non-stationary configuration since $|\Delta| = d + 1$. The free-form term $f_\theta$ computes one coefficient per vertex and attribute and therefore has an output dimension of $(d + 1) \cdot m$ whereas the transport term estimates a velocity vector for each cell and attribute resulting in an output dimension of $m \cdot d$. The number of hidden layers and other configuration details for each experiment are listed in Appendix D. We use `scikit-fem` (Gustafsson & McBain, 2020) to compute the inner products between basis functions over the mesh cells and the dopri5 solver in `torchdiffeq` (Chen et al., 2018) to solve the resulting ODE. See Appendix E for details on model training.

## 4 EXPERIMENTS

**Baselines.** We evaluate FEN and T-FEN against a selection of models representing a variety of temporal prediction mechanisms: Graph WaveNet (GWN) combines temporal and graph convolutions (Wu et al., 2019); Physics-aware Difference Graph Network (PA-DGN) estimates spatial derivatives as additional features for a recurrent graph network (Seo et al., 2020); the continuous-time MPNN-based model by (Iakovl et al., 2021) which we will call CT-MPNN, uses a general MPNN to learn the continuous-time dynamics of the data.[1] See Appendix D for the configuration details.

GWN and PA-DGN use 12 and 5 timesteps respectively as input while the ODE-based models make their predictions based on a single timestep. To ensure a fair comparison, we do not evaluate any of the models on the first 12 timesteps of the test sets.

**Datasets.** For our experiments we have chosen two real-world datasets and a synthetic one. The Black Sea dataset provides data on daily mean sea surface temperature and water flow velocities on the Black Sea over several years. ScalarFlow is a collection of 104 reconstructions of smoke plumes from 2 seconds long, multi-view camera recordings (Eckert et al., 2019). In each recording a fog machine releases fog over a heating element in the center of the domain which then rises up with the hot air through convection and buoyancy. CylinderFlow contains simulated fluid flows around a cylinder along a channel simulated with the inviscid Navier-Stokes equations (Pfaff et al., 2021).

All our experiments are run on hold-out datasets, i.e. the year 2019 of the Black Sea data and the last 20 recordings of ScalarFlow. See Appendix C for a detailed description of the datasets, our sparse subsampling procedure, choice of normalization, and train-validation-test splits.

Table 1: Mean absolute error and number of function evaluations for multi-step forecasting on the test set. We forecast 10 steps ahead and averaged the metrics over 10 runs. Results marked with $*$ include models that could not be trained to completion because of excessive memory requirements.

| | ScalarFlow | | Black Sea | | CylinderFlow | |
|---|---|---|---|---|---|---|
| | MAE $\times 10^{-2}$ | NFE | MAE $\times 10^{-1}$ | NFE | MAE $\times 10^{-2}$ | NFE |
| PA-DGN | $13.86 \pm 0.14$ | - | $9.20 \pm 0.05$ | - | $5.46 \pm 0.02$ | - |
| GWN | $9.79 \pm 0.05$ | - | $8.94 \pm 0.05$ | - | $\mathbf{3.04 \pm 0.28}$ | - |
| CT-MPNN | $\mathbf{8.97 \pm 0.06}$ | $157.2 \pm 24.4$ | $\mathbf{8.54 \pm 0.05}$ | $181.6 \pm 32.9$ | $5.18 \pm 0.43*$ | $42.2 \pm 42.5$ |
| FEN | $9.14 \pm 0.05$ | $225.0 \pm 43.0$ | $8.78 \pm 0.11$ | $102.3 \pm 27.6$ | $\mathbf{2.87 \pm 0.17}$ | $123.6 \pm 33.1$ |
| T-FEN | $\mathbf{9.04 \pm 0.06}$ | $315.5 \pm 165.0$ | $8.72 \pm 0.05$ | $65.4 \pm 4.5$ | $4.09 \pm 0.62*$ | $67.1 \pm 22.4$ |

**Multi-step Forecasting.** The main objective of our models as well as the baselines are accurate forecasts and we train them to minimize the mean absolute error over 10 prediction steps. We chose this time horizon because it is long enough that a model needs to learn non-trivial dynamics to perform well and more accurate dynamics lead to better performance. At the same time it is short enough that not too much uncertainty accumulates. For example, if we train any of the models on Black Sea on much longer training sequences, they learn only diffusive smoothing because the correlation between the temperature distribution today and in 30 days is basically nil, so that the only winning move is not too play – similar to how forecasting the weather for more than 10 days is basically impossible (Mailier, 2010).

For Table 1 we have evaluated the models on all possible subsequences of length 10 of the respective test set of the ScalarFlow and Black Sea datasets and on all consecutive subsequences of length 10 in the CylinderFlow test set because of the size of the dataset. The results show that FEN is competitive with the strongest baselines on all datasets. On real-world data, the transport component in T-FEN provides a further improvement over the base FEN model. The number of function evaluations for all ODE-based models depend strongly on the type of data. While CT-MPNN is the cheapest model on ScalarFlow in terms of function evaluations by the ODE solver, FEN and T-FEN are more efficient on Black Sea. On the synthetic CylinderFlow dataset, neither CT-MPNN nor T-FEN can by trained fully because the learned dynamics quickly become too costly to evaluate. These results show that including prior knowledge on the form of the dynamics $F$ in the model can improve the prediction error when the additional assumptions hold.

---

[1]GWN and PA-DGN condition their predictions on multiple timesteps, giving them a possible advantage over ODE-based models since they can use the additional input for denoising or to infer latent temporal information.

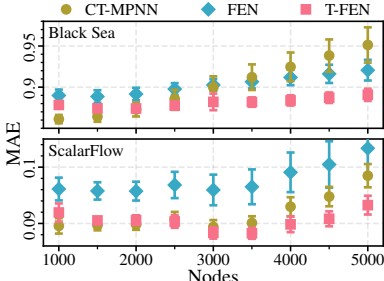

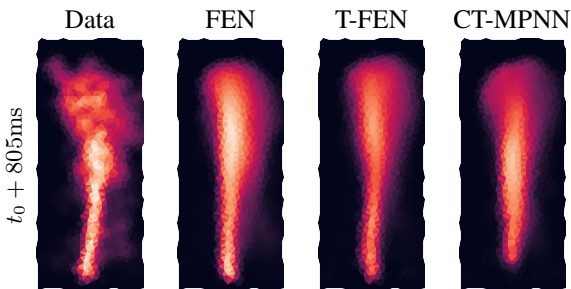

Figure 3: Predictive accuracy of models trained on 1000 nodes and evaluated on increasingly fine subsamplings of the test data.

Figure 4: Evolving the learned dynamics of FEN models and the strongest baseline trained on 10-step forecasting forward for 60 time steps reveals the differences between these models to the human eye.

**Super-Resolution.** Graph-based spatio-temporal forecasting models make predictions on all nodes jointly. Training these models on many points or fine meshes therefore requires a prohibitive amount of memory to store the data points and large backwards graphs for backpropagation. One way around this problem is to train on coarse meshes and later evaluate the models on the full data. In Fig. 3 we compare FEN and T-FEN against the strongest baseline on the Black Sea dataset. While all three models increase in prediction error as the mesh resolution increases, FEN models deteriorate at a markedly slower rate than CT-MPNN. We ascribe this to the fact that in FEN and T-FEN estimation of the dynamics and its effect on the nodes are to a certain extent separated. The terms $f_\theta$ and $g_\theta$ estimate the dynamics on each cell, but the effect on the node values is then controlled by the inner products of basis functions which incorporate the shape of the mesh cells in a theoretically sound way. In CT-MPNN on the other hand, the model learns these aspects jointly which makes it more difficult to generalize to different mesh structures. See Appendix F for the complete results on both datasets and a visual comparison of low and high resolution data.

**Extrapolation.** While we train the models on 10-step forecasting, it is instructive to compare their predictions on longer time horizons. Fig. 4 shows the predictions of the three strongest models on a smoke plume after 60 steps. Because these models are all ODE-based, they were conditioned on a single timestep and have been running for hundreds of solver steps at this point. First, we notice that all three models have the fog rise at about the same, correct speed. However, with CT-MPNN the fog disappears at the bottom because the model does not learn to represent the fog inlet as a fixed, local source as opposed to our models. Comparing FEN and T-FEN, we see that the former's dynamics also increase the density further up in the fog column where no physical sources exist, while the latter with its separate transport term modeling convection keeps the density stable also over longer periods of time. See Appendix G for a comparison with all baselines at multiple time steps.

**Interpretability.** In FEN the free-form term $f_\theta$ models the dynamics $F$ as a black box and as such its activations are equally opaque as the dynamics we derived it from. However, as we have seen in Section 3.1, by making assumptions on the structure of $F$ we impose structure onto the model and this structure in turn assigns meaning to the model's components. For a T-FEN we see that the transport term corresponds to a velocity field for each attribute, while the free-form term absorbs the remainder of the dynamics. Fig. 5 shows that this disentanglement of dynamics works in practice. In the ScalarFlow data, we have a fog inlet at the bottom, i.e. a source of new energy, and from there the fog rises up by convection. Plotting the contributions of $f_\theta$ and $g_\theta$ to $\partial_t \boldsymbol{y}$ separately shows that the free-form term represents the inlet while the transport term is most active in the fog above. Instead of inspecting the contributions to $\partial_t \boldsymbol{y}$, we can also investigate the learned parameter estimates directly. Fig. 6 shows that the transport term learns a smooth velocity field for the sea surface temperature on the Black Sea.

## 5 RELATED WORK

**Spatio-Temporal Forecasting.** Various DNNs, and in particular GNNs, have been designed to model complex spatial and temporal correlations in data, with applications ranging from forecasting traffic

Free-Form   Transport

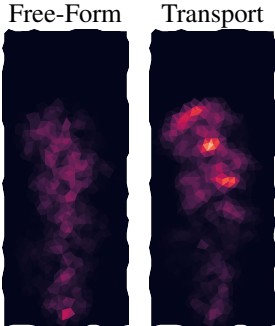

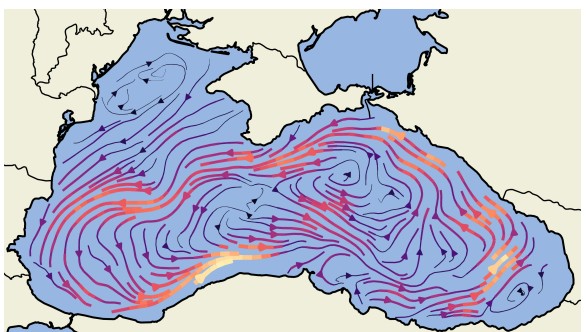

Figure 5: Contribution to $\partial_t \boldsymbol{y}$ from the free-form and transport term in a T-FEN on a snapshot of ScalarFlow.

Figure 6: Learned temperature flow field in a T-FEN after 10 steps on Black Sea data. The model recognized the physical relationship between the features.

flow (see (Jiang & Luo, 2021) and (Yin et al., 2021) for a comprehensive survey), to forecasting COVID-19 (Kapoor et al., 2020) and sea-surface temperature (de Bézenac et al., 2018). For a detailed review of both classical and deep learning approaches, which can be roughly categorized as based on point-cloud trajectories, regular grids, or irregular grids, see Shi & Yeung (2018). As one representative of GNN-based methods, which tend to be SOTA, we compare FENs with GWN (Wu et al., 2019). GWN uses a combination of graph convolution and 1D dilated temporal convolutions to learn spatio-temporal properties of traffic networks.

**Neural Networks and Differential Equations.** In 1998 Lagaris et al. (1998) introduced the first neural network approach to solve ODEs and PDEs. They formulate the optimization goal as fulfilling the properties of the differential equation. Raissi et al. (2019) extend this idea and propose a framework to respect any given general nonlinear PDE for continuous- and discrete-time models. Another neural approach to learn PDEs from data is PDE-Net (Long et al., 2018). Its two main components are: learning the differential operators of PDEs through convolutional kernels and applying a neural network to approximate the nonlinear response functions. Building on that work, PDE-Net 2.0 (Long et al., 2019) uses a symbolic network to approximate the nonlinear response function. de Bézenac et al. (2018) derive a model from the convection-diffusion equation and estimate the flow field on a regular grid with a CNN. Chen et al. (2018) leverage ODE solvers in their neural architecture to create a model family of continuous depth called neural ODEs. Ayed et al. (2019) build upon neural ODEs to make spatio-temporal forecasts by learning the dynamics of all data points jointly with the structure of the domain encoded via the node positions as additional features. GNODE (Poli et al., 2019) brings the framework of neural ODEs of Chen et al. (2018) into the graph domain and provides continuous depth GNNs to make traffic predictions. In dynamical systems the variable depth component coincides with the time dimension which allows the network to work with observations at varying timesteps. Iakovl et al. (2021) extend GNODE with distance vectors between nodes as edge features and use it for PDE prediction. Mitusch et al. (2021) derive a model from FEM similar to our work but integrate the learned dynamics against the basis functions numerically and disregard the mesh to avoid designing the discretization into the model.

## 6   CONCLUSION

We have introduced Finite Element Networks as a new spatio-temporal forecasting model based on the established FEM for the solution of PDEs. In particular, we have shown how FENs can incorporate prior knowledge about the dynamics of the data into their structure through assumptions on the form of the underlying PDE and derived the specialized model T-FEN for data with convection dynamics. Our experiments have shown that FEN models are competitive with the strongest baseline model from three spatio-temporal forecasting model classes on short-term forecasting of sea surface temperature and gas flow and generalize better to higher resolution meshes at test time. The transport component of T-FEN boosts the performance and additionally allows meaningful introspection of the model. Its structure is directly derived from the assumed form of the underlying PDE and as such the transport and free-form terms disentangle the dynamics into a convection component and the remainder such as inlets and sinks.

## ACKNOWLEDGEMENTS

We thank Leon Hetzel for helpful discussions and suggestions.

This research was supported by the Bavarian State Ministry for Science and the Arts within the framework of the Geothermal Alliance Bavaria project.

## ETHICS STATEMENT

We propose a general model for spatio-temporal forecasting which can be applied in a variety of settings. Among these are traffic and crowd flow predictions which have the potential for abuse. However, due to the close connection to PDEs, we see it primarily as a model for physical systems in earth sciences and engineering, with impacts in the study of climate change and understanding of complex systems. Finally, we can also imagine its application in modeling the spread of infectious diseases which also employs PDEs today (Viguerie et al., 2021).

## REPRODUCIBILITY

To maximize the reproducibility of our experimental results, we have fixed seeds for every domain subsampling, training run and evaluation and recorded these in configuration files, so that everything can be re-run. We obtained all seed values from `numpy.random.SeedSequence` to guarantee that the streams of random numbers from the random number generators (RNGs) are independent and have high sample quality. While we fixed all seeds, model training is still non-deterministic because the message passing step in graph-based models relies on a scatter operation. Scattering is implemented via atomic operations on GPUs, which can re-order floating point operations such as additions between runs with the same seed inducing non-determinism (Fey & Lenssen, 2019). Nonetheless, we have observed that multiple runs from the same seed only diverge slowly as training progresses and the variability in performance between runs from different seeds is small for all models anyway as can be seen in the standard deviation of the mean absolute error in Table 1.

We have used an NVIDIA GeForce GTX 1080 Ti for all experiments and evaluations.

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

## A    DISCUSSION

We have presented a way to derive a spatio-temporal forecasting model directly from FEM. In addition to establishing a connection between this well-known theory behind many scientific simulations and neural ODE models, our derivation opens a way to include knowledge in the form of PDEs directly into the structure of a machine learning model, as we have shown by the example of the convection equation and T-FEN Extensions of the model with other first or second order, linear PDEs such as the heat equation or the wave equation should follow with a similar derivation. However, our approach cannot be applied directly when derivatives of higher order are involved.

The problem fundamentally stems from our choice of basis functions. The P1 basis is piecewise linear and thus has vanishing second and higher order derivatives. This means that, at the end of the model derivation, any basis function involved in fixed weights such as $\langle \varphi^{(j)}, \varphi^{(i)} \rangle$ can appear with at most a first derivative, otherwise the expression will evaluate to 0. A solution would lie in extending our approach to higher order basis functions, though this is a non-trivial endeavor. In such a setting, the values at the nodes of a triangular mesh would no longer suffice to determine all coefficients in the new basis. In future research, we will work on constraining these new coefficients and make learning with higher-order basis functions a well-posed learning problem.

## B    DERIVATION OF THE TRANSPORT TERM

Let $F$ be just a convection term.

$$F\left(t, \boldsymbol{x}, u, ...\right)_k = -\nabla \cdot \left(v^{(k)}(t, \boldsymbol{x}, u, ...)u_k\right) \tag{23}$$

Because of the linearity of the inner product $\langle \cdot, \cdot \rangle_\Omega$, we can ignore other additive components of $F$ in this derivation. In the rest of this section, we will write $v_k$ for $v^{(k)}(t, \boldsymbol{x}, u, ...)$. We begin by plugging $F$ into the message matrix from Eq. (14).

$$M_{ik} = -\langle \nabla \cdot \left(v_k u_k\right), \varphi^{(i)} \rangle_\Omega \tag{24}$$

Next, we split the domain into the mesh cells.

$$= -\sum\nolimits_{\Delta \in \mathcal{T}_i} \langle \nabla \cdot \left(v_k u_k\right), \varphi^{(i)} \rangle_{\mathrm{CH}(\Delta)} \tag{25}$$

Now we apply the product rule.

$$= -\sum\nolimits_{\Delta \in \mathcal{T}_i} \langle v_k \cdot \nabla u_k + \left(\nabla \cdot v_k\right) u_k, \varphi^{(i)} \rangle_{\mathrm{CH}(\Delta)} \tag{26}$$

We assume that the velocity field is divergence-free, i.e. $\nabla \cdot v_k = 0$.

$$= -\sum\nolimits_{\Delta \in \mathcal{T}_i} \langle v_k \cdot \nabla u_k, \varphi^{(i)} \rangle_{\mathrm{CH}(\Delta)} \tag{27}$$

Now we expand $u$ in the P1 basis (remember that we have encoded the data $\boldsymbol{y}$ in the coefficients of $u$)

$$= -\sum\nolimits_{\Delta \in \mathcal{T}_i} \langle v_k \cdot \nabla \sum\nolimits_{j=1}^{N} y_k^{(j)} \varphi^{(j)}, \varphi^{(i)} \rangle_{\mathrm{CH}(\Delta)} \tag{28}$$

and make use of the linearity of the inner product and gradient once more.

$$= -\sum\nolimits_{\Delta \in \mathcal{T}_i} \sum\nolimits_{j=1}^{N} y_k^{(j)} \langle v_k \cdot \nabla \varphi^{(j)}, \varphi^{(i)} \rangle_{\mathrm{CH}(\Delta)} \tag{29}$$

Because the inner product is restricted to $\Delta$, we can restrict the inner sum to vertices of $\Delta$.

$$= -\sum\nolimits_{\Delta \in \mathcal{T}_i} \sum\nolimits_{\boldsymbol{x}^{(j)} \in \Delta} y_k^{(j)} \langle v_k \cdot \nabla \varphi^{(j)}, \varphi^{(i)} \rangle_{\mathrm{CH}(\Delta)} \tag{30}$$

Finally, we factorize the inner products in the same way as we did it for the free-form term

$$= -\sum\nolimits_{\Delta \in \mathcal{T}_i} \sum\nolimits_{\boldsymbol{x}^{(j)} \in \Delta} y_k^{(j)} v_k \cdot \langle \nabla \varphi^{(j)}, \varphi^{(i)} \rangle_{\mathrm{CH}(\Delta)} \tag{31}$$

and plug in the neural network $g_\theta$ approximating the velocity field.

$$= -\sum\nolimits_{\Delta \in \mathcal{T}_i} \sum\nolimits_{\boldsymbol{x}^{(j)} \in \Delta} y_k^{(j)} g_{\theta, \Delta, k}^{(t,i)} \cdot \langle \nabla \varphi^{(j)}, \varphi^{(i)} \rangle_{\mathrm{CH}(\Delta)} \tag{32}$$

If we stack this expression over all features $k$ and translate the result into the MPNN framework, we arrive at the transport term that we present as part of T-FEN in Section 3.1.

## C DATASETS

In this section, we give a brief overview over each of the datasets we used in Section 4 and how we pre-processed them.

**Subsampling.** Both real-world datasets that we use provide large amounts of dense data on a 2D or 3D grid, which we subsample to simulate sparse measurements of a physical system and make the amount of data per timestep manageable. A non-trivial problem is the selection of sampling points and we decided to use the k-Medoids algorithm. This way we ensure that we use actual data points as sampling points and reach a roughly uniform but still random cover of the domain. A uniform covering is especially important for the Black Sea dataset because of the non-convex shape of the domain and small bays on the border that we still want to cover.

### C.1 CYLINDERFLOW

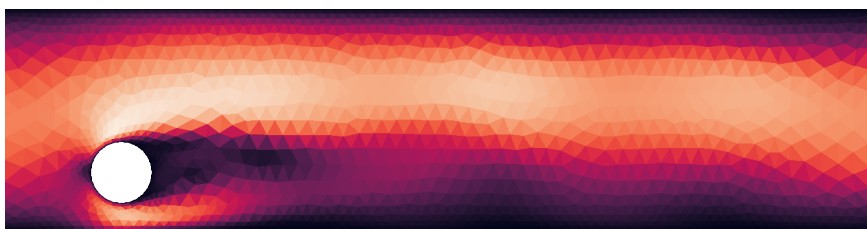

(a) The magnitude of the velocity field

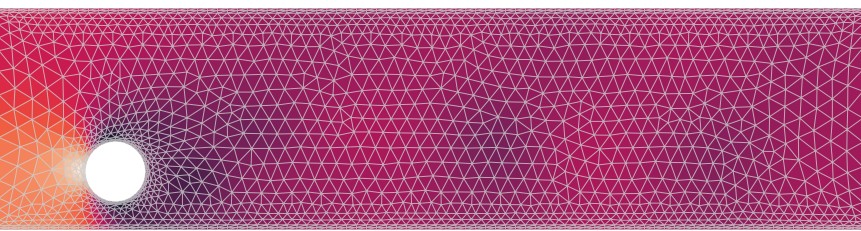

(b) The pressure field with the simulation mesh overlaid

Figure 7: A sample from the CylinderFlow dataset. The upper and lower boundaries are impenetrable walls. The left-hand boundary is an inflow with a constant velocity profile over time. The mesh resolution increases towards the boundaries at the top and bottom as well as in the immediate vicinity of the cylinder.

The CylinderFlow dataset is a collection of simulated flows characterized by velocity and pressure fields released by Pfaff et al. (2021). The domain is a two-dimensional channel with impenetrable boundaries at the top and bottom and a circular obstacle of varying size and position that blocks the flow. On the left-hand boundary, the velocity field is fixed to an inflow profile that is constant over time but varies between samples. Depending on the obstacle, inflow profile, and flow parameters the resulting velocity and pressure fields exhibit distinct behaviors, varying between stable flows and oscillations. Pfaff et al. used the COMSOL solver to generate the data by solving the incompressible Navier-Stokes equations. See Fig. 7 for a visualization of the velocity and pressure fields as well as the simulation mesh.

The training set contains 1000 simulations, while validation and test set have 100 each. Every sequence consists of 600 steps of the flow sampled at a time resolution of $\Delta t = 0.01$s. Spatially, the data is discretized on meshes with about 1800 nodes. A special property of this dataset is that each sequence has a different mesh, because each mesh has a notch representing the cylinder obstacle which varies in size and position.

To enforce the boundary conditions, we fix the velocity fields for all methods on the boundaries except for the outflow, similar to Pfaff et al.. On the boundaries of the channel as well as the cylinder boundary, we fix the velocities to zero. The velocity inflow profile stays constant over time and we

hold it fixed during training and inference. The pressure, however, is unconstrained throughout the domain and can also vary on the boundaries.

Due to the size of the dataset, we select only the first 100 sequences for training and train only on subsequences starting at a multiple of 10 steps, in effect eliminating overlapping training sequences and reducing the size of the training dataset by another factor of 10. For the evaluation, we also choose the subsequences to test on in the same way, so every subsequence of length 10 starting at time step 0, 10 and so on.

We normalize each feature by its mean and standard deviation across all training sequences.

## C.2    BLACK SEA

The Black Sea data is a reanalysis dataset of daily and monthly mean fields in the Black Sea of variables such as temperature, flow velocity, and salinity and 31 depth levels from 01/01/1992. The whole dataset is available as the BLKSEA_MULTIYEAR_PHY_007_004[2] product from the EU Copernicus Marine Service under a custom license[3] granting permission to freely use the data in research under the condition of crediting the Copernicus Marine Environment Monitoring Service.

The spatial resolution of the raw data is 1/27° × 1/36° on a regular mesh of size 395×215 covering the Black Sea. Of the resulting 84,925 grid points, about 40,000 contain actual sea data and we subsample these valid points to create our sparse datasets. We mesh the chosen sample points with the Delaunay algorithm and filter acute mesh cells on the boundary as described in Appendix H. Additionally, we remove mesh cells that cover more land than water to avoid connecting nodes that are close by air distance but far apart through the sea.

We use daily data from 01/01/2012 to 31/12/2017 for training, 01/01/2018 to 31/12/2018 for validation and 01/01/2019 to 31/12/2019 for testing. As features we choose the zonal velocities and the water temperature. Both features are taken at a depth of 12.54m because the dynamics at the surface are strongly influenced by wind, sun, and cloud cover, which are not part of the data and thus make the dynamics non-deterministic from the perspective of the model.

On this dataset, we normalize the node positions as well as the features. The node positions are just normalized by their mean and average standard deviation in latitudinal and longitudinal direction over the training set as are the zonal velocity features. For the temperature, we chose mean and standard deviation grouped by calendar day over the training date range for the normalization, e.g. all January 1sts over the years 2012 to 2017 are grouped together, because the temperature exhibits a clear yearly cyclicity.

## C.3    SCALARFLOW

ScalarFlow is a dataset of 104 smoke plumes published by Eckert et al. (2019) under the Creative Commons license CC-BY-NC-SA 4.0 (Eckert et al., 2019). Eckert et al. created a controlled environment in which fog from a fog machine rises in an air column over a heating element. They captured the resulting flow on video with a calibrated multi-camera setup and reconstructed the velocity and density fields of each recording on a dense 100×178×100 grid for 150 timesteps with their proposed method. At each grid point, the dataset provides 3-dimensional velocity vectors and the current fog density. The resulting dataset represents high-resolution measurements of a dynamical physical system that is mostly driven by buoyancy and transport processes but also evokes turbulence. All plots of this dataset show the fog density.

Of the 104 recordings we use the first 64 for training, the next 20 for validation and the final 20 as a test set. Before subsampling the grid points, we reduce the data to 2D by averaging over the depth (Kohl et al., 2020) and restrict the data to the central 60 points in $x$-direction and bottom 140 points in $y$-direction, because the fog does not reach points outside of that central box in almost any of the recordings. We normalize the coordinates of the sample points by their mean and standard deviation as well as the features.

---

[2]https://resources.marine.copernicus.eu/product-detail/BLKSEA_MULTIYEAR_PHY_007_004

[3]https://marine.copernicus.eu/user-corner/service-commitments-and-licence

# D    MODEL CONFIGURATIONS

We trained all models on all datasets for at most 50 epochs or 24 hours. Most trainings have been stopped early after no improvement in the validation score has been observed for 5 epochs or the ODE-based models ran out of memory due to excessive memory requirements, when the learned dynamics required too many function evaluations to solve.

The three datasets present us with different circumstances regarding the basic properties of optimal dynamics that would model them well. Both Black Sea and ScalarFlow are best served by a non-stationary model, because it is reasonable to assume that the dynamics of the sea depend on the position and ScalarFlow has an inlet at the bottom with fixed position. On top of that, we further expect that the BlackSea dataset has a time dependence exceeding the global, yearly fluctuations that we subtract with our normalization scheme. Therefore, we make all models non-stationary and non-autonomous on Black Sea and just non-stationary on ScalarFlow. On CylinderFlow, the models are autonomous and stationary.

The mechanism to make FEN and T-FEN non-stationary and non-autonomous is described in Section 3.1. For the baselines, we concatenate the time stamps and the node positions respectively to the node features.

ODE-based models such as FEN and CT-MPNN require many function evaluations when their dynamics change abruptly. One possible trigger for that can be a jump in their inputs. This would, for example, occur if we would represent the time of the year on Black Sea as a number between 0 and 1 to capture the yearly cyclicity of the dynamics. Then we would get a discontinuity in the input at the turn of the year. To avoid this jump, we instead embed the time feature on Black Sea two-dimensionally as $(\sin(\tilde{t}), \cos(\tilde{t}))$ where $\tilde{t}$ is a map from the current time within a year to $[-1, 1]$.

For GWN, we use a batch size 6 on all datasets and a batch size of 3 with PA-DGN. Due to an implementation detail of `torchdiffeq` as of the submission of this paper, batched solving of ODEs with adaptive-step solvers can introduce interference between independent ODEs, because internal error estimates in the solver are computed across ODEs and the dynamics are stepped jointly. Therefore, we train and evaluate the ODE-based models with a batch size of 1.

## D.1    FEN & T-FEN

Both $f_\theta$ and $g_\theta$ are MLPs with 4 hidden layers and `tanh` non-linearities. In FEN the layers of $f_\theta$ have a width of 128. For T-FEN we chose to reduce the width of the hidden layers of both $f_\theta$ and $g_\theta$ to 96, so that both FEN and T-FEN have a comparable number of parameters and we can trace back any difference in performance to difference in model structure and not capacity. See Table 2 for the parameter counts. We solve the feature dynamics with a `dopri5` solver with an absolute tolerance and relative tolerance of $10^{-6}$ on CylinderFlow and Black Sea, and $10^{-7}$ on ScalarFlow. On initialization, we set the last weights and bias of the last layers of both MLPs to 0 to start with the constant dynamics as a reasonable initialization for training. For parameter learning, we use the Adam optimizer with a learning rate of $10^{-3}$ (Kingma & Ba, 2015).

## D.2    BASELINES

**CT-MPNN.** Following Iakovl et al. (2021), our CT-MPNN implementation models both the message function and the update function as MLPs with `tanh` non-linearities. Contrary to the default configuration used by Iakovl et al. (2021), we increased the widths of the MLPs from 60 to 128 and the message dimension from 40 to 128 to make the model capacity comparable to the other baselines. Instead of an implicit Adams-Bashforth solver we used the same `dopri5` solver as for FEN and T-FEN with an absolute tolerance of $10^{-6}$ and a relative tolerance of $10^{-7}$. This both increased the models performance and avoided frequent convergence issues in the step function of the Adams-Bashforth solver. Finally, we exchanged the RPROP opti-

Table 2: Parameter counts for all models.

|  | Black Sea | ScalarFlow |
|---|---|---|
| PA-DGN | 269140 | 269140 |
| GWN | 459750 | 466160 |
| CT-MPNN | 134403 | 134532 |
| FEN | 53129 | 53772 |
| T-FEN | 60975 | 61844 |

mizer with a standard Adam optimizer and we compute the gradients the same in FEN and T-FEN by backpropagation through the ODE solver.

We apply the same zero-initialization and increasing training sequence lengths, that we use for our models, to this baseline.

**PA-DGN.** For PA-DGN we choose two graph network layers of two layers each with a hidden size of 64. We condition the model on 5 timesteps. To represent the structure of the domain, we pass the 4-nearest neighbor graph of the nodes to the model following authors of the method (Seo et al., 2020).

**GWN.** For GWN we use the improved configuration described by Shleifer et al. (2019) with 40 convolution channels, extra skip connections and four of GWN's WaveNet inspired blocks with two layers each. The model is conditioned on 12 time steps of data and predicts 10 steps ahead. For the long-term extrapolation in Appendix G, we apply the model auto-regressively to get predictions for more than 10 steps. We do not learn an adaptive adjacency matrix because that comes with a quadratic memory cost in the number of nodes and Wu et al. have shown that GWN only performs 1.2 % worse on METR-LA without the adaptive adjacency matrix. For the computation of the forward and backward transition matrices, we pass in the adjacency matrix derived from the mesh of the domain where the edges are weighted by the Gaussian kernel

$$e(\boldsymbol{x}^{(i)}, \boldsymbol{x}^{(j)}) = \exp\left(-\frac{\left\|\boldsymbol{x}^{(i)} - \boldsymbol{x}^{(j)}\right\|_2^2}{\sigma^2}\right) \tag{33}$$

where $\sigma$ is the standard deviation of $\left\|\boldsymbol{x}^{(i)} - \boldsymbol{x}^{(j)}\right\|_2$.

## E  TRAINING

We train our models with the Adam optimizer (Kingma & Ba, 2015) by minimizing the multi-step prediction error in terms of the mean $L_1$ distance between predicted and actual node features

$$\mathcal{L}(\hat{\boldsymbol{y}}, \boldsymbol{y}) = \frac{1}{NT} \sum_{i=1}^{N} \sum_{j=1}^{T} \|\hat{\boldsymbol{y}}^{(t_j, i)} - \boldsymbol{y}^{(t_j, i)}\|_1 \tag{34}$$

where $\hat{\boldsymbol{y}}^{(t_j, \cdot)}$ is the prediction at time $t_j$. To accumulate gradients, we backpropagate through the operations of the ODE solver, also known as discretize-then-optimize, as it improves training time significantly compared to backpropagation with the adjoint equation (Onken & Ruthotto, 2020).

We observed that training our models to predict complete sequences from the beginning can get them stuck in local minima that take many epochs to escape. To stabilize training, we begin by training on subsequences of length $s = 3$ and iteratively increase $s$ by 1 per epoch until it reaches the maximum training sequence length.

## F  SUPER-RESOLUTION

For our super-resolution experiment, we have subsampled both datasets 3 times for each fixed number of nodes and evaluated each of the 10 trained models per model class against each of them. Therefore, the means (denoted by the markers) and standard deviations (denoted by the error bars) in Fig. 3 and Fig. 8 have been computed over 30 evaluations.

Inspecting the full results in Fig. 8 reveals two things. While T-FEN performs best in super-resolution on both datasets, GWN and PA-DGN also only deteriorate slightly on finer meshes. However, this is offset by the fact that these models did not achieve such a good fit to begin with and are still out-performed by T-FEN on every resolution. CT-MPNN achieved a lower prediction error than T-FEN on the original mesh granularity but generalizes markedly worse than both FEN and T-FEN to finer meshes. We suppose that this is due to the fact that CT-MPNN uses a general MPNN internally, while FEN and T-FEN have an inductive bias towards producing physically sensible predictions because of their structural connection to FEM and PDEs.

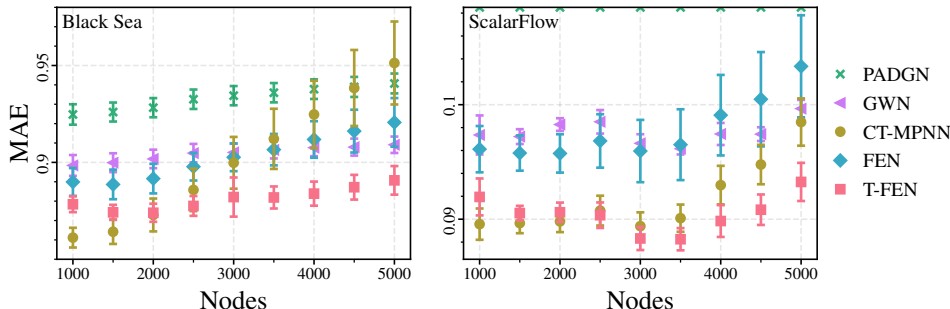

Figure 8: All models were trained on 1000 node subsamples of the data for 10-step prediction and then evaluated on the same task but on increasingly fine meshings of the domain.

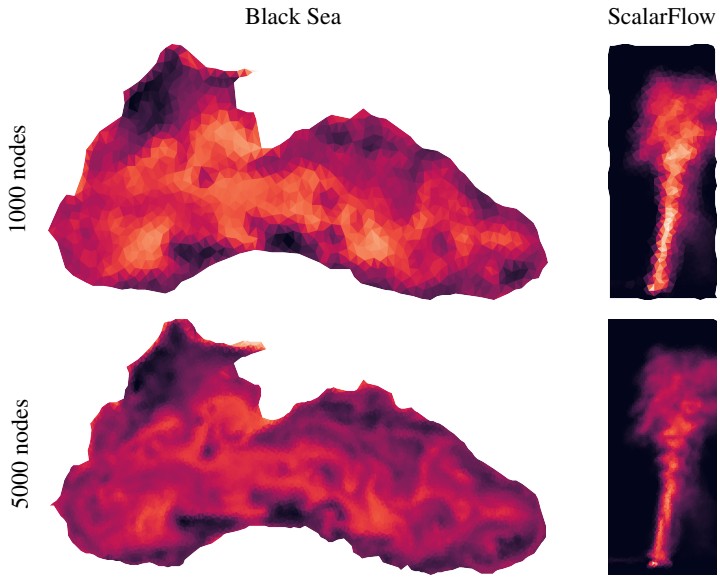

Figure 9: A visual comparison of the data when subsampled at 1000 nodes and at 5000 nodes.

## G  LONG TERM EXTRAPOLATION

In Fig. 10 we show 60-step extrapolations of the all models trained on 10-step forecasting. The first thing to note is that PA-DGN struggles with the directionality inherent in the ScalarFlow dataset. While its prediction evolves over time, it exhibits a mostly diffusive dynamic.

From a visual standpoint, GWN produces the most realistic looking forecasts with small-scale details even after many timesteps, especially compared to the ODE-based models that beat GWN in short-term forecasting. We see two reasons for that. First, ODE-based models that learn the data dynamics directly, i.e. all three of FEN, T-FEN, and CT-MPNN, have to work with a tight information bottleneck. Because these models carry information through time only in the features themselves and have no latent state, they can condition their prediction only on a single time step and this bottleneck makes it difficult to conserve fine details over time. GWN on the other hand predicts 10 timesteps at a time and conditions each forecast on the past 12 timesteps. In this way, GWN can extract more information from the input data and also carry that information forward 10 steps at a time whereas ODE-based models have to conserve the information through roughly 100 solver steps to reach $t_{10}$, see the number of function evaluations in Table 1.

Second, FEN, T-FEN, and CT-MPNN are ODE models and as such are biased towards smooth predictions over time and over long time frames this leads to a smoothing in space. Yet, on short time frames, before smoothing sets in, these models are able to model these physical processes more

accurately, because ODEs are natural models for these data. For a discussion of the differences between the FEN, T-FEN, and CT-MPNN predictions, see Section 4.

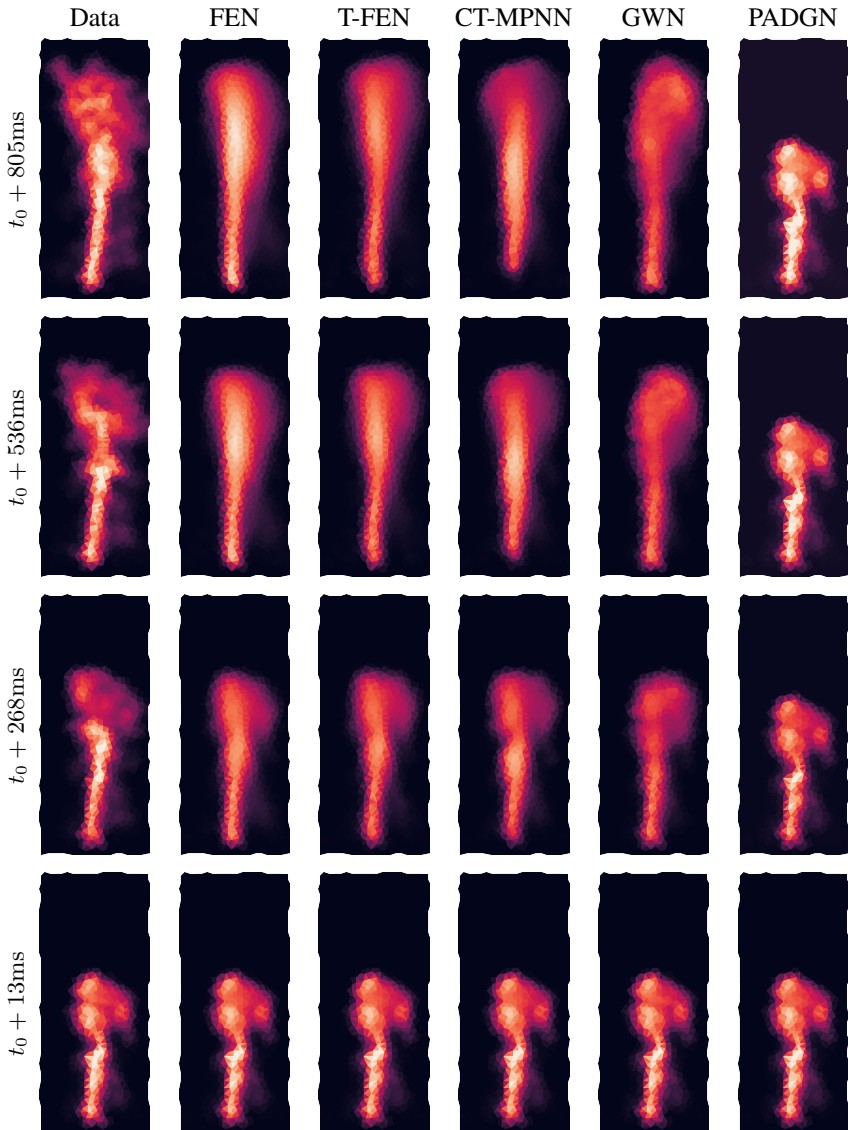

Figure 10: Predictions of all models for a rising smoke plume after 1, 20, 40 and 60 time steps.

## H  FILTERING HIGHLY ACUTE CELLS ON THE BOUNDARY

Delaunay triangulation of a set of points can produce very long cells on the boundary as in Fig. 11 when several points close to the boundary are almost co-linear. This can just occur in the data but it can also happen during generation of synthetic data. If some boundary nodes in synthetic data are exactly co-linear such as in grid domains, they might lose the exact co-linearity due to the inexactness of floating point numbers and operations if we, for example, rotate the domain. To ensure that these artifacts do not impact our models, we pre-process the triangulation with Algorithm 1 and an angle threshold of $\varepsilon = \frac{10\pi}{180}$.

---

**Algorithm 1** Filtering cells with small interior boundary-adjacent angles from a triangulation given a threshold angle and a set of $d - 1$ dimensional boundary faces

---

1: **function** FILTERCELLS(triangulation $\mathcal{T}$, threshold $\varepsilon > 0$, boundary faces $\mathcal{B}$)
2:      **for all** $\Delta \in \mathcal{T}$ **do**            $\triangleright$ Count number of faces on the boundary for each cell
3:          $c_\Delta \leftarrow |\{\tilde{\Delta} \mid \tilde{\Delta} \subset \Delta, |\tilde{\Delta}| = d, \tilde{\Delta} \in \mathcal{B}\}|$
4:      $\mathcal{F} \leftarrow \{\Delta \mid \Delta \in \mathcal{T}, c_\Delta = 1\}$
5:      **while** $|\mathcal{F}| > 0$ **do**
6:          $\Delta \leftarrow \text{pop}(\mathcal{F})$
7:          **if** $c_\Delta \neq 1$ **then**            $\triangleright$ Check that $\Delta$ is not at an edge or corner of the domain
8:              **continue**
9:          $(\tilde{\Delta}, \xi) \leftarrow \text{split}(\Delta)$            $\triangleright$ Split boundary face and interior node
10:          **if** MINBOUNDARYANGLE($\tilde{\Delta}, \xi$) $< \varepsilon$ **then**
11:              $\mathcal{T} \leftarrow \mathcal{T} \setminus \{\Delta\}$            $\triangleright$ Remove cell
12:              **for all** $\hat{\Delta} \subset \tilde{\Delta} \mid |\hat{\Delta}| = d - 1$ **do**        $\triangleright$ Add newly become boundary faces to $\mathcal{B}$
13:                  $\mathcal{B} \leftarrow \mathcal{B} \cup \{\hat{\Delta} \cup \{\xi\}\}$
14:                  **for all** $\Delta' \in \mathcal{T} \mid \hat{\Delta} \subset \Delta', \xi \in \Delta'$ **do**
15:                      $c_{\Delta'} \leftarrow c_{\Delta'} + 1$
16:                      $\mathcal{F} \leftarrow \mathcal{F} \cup \{\Delta'\}$        $\triangleright$ Add cells that are now boundary cells to $\mathcal{F}$
17:      **return** $\mathcal{T}$

18: **function** MINBOUNDARYANGLE($\tilde{\Delta}, \xi$)
19:      $B \leftarrow \text{project}(\xi, \tilde{\Delta})$            $\triangleright$ Project $\xi$ onto the hyperplane given by $\tilde{\Delta}$
20:      $\gamma \leftarrow \pi$
21:      **for all** $A \in \tilde{\Delta}$ **do**
22:          $\gamma \leftarrow \min(\gamma, \text{angle}(AB, A\xi)$      $\triangleright$ Find the angle between the line segments $AB$ and $A\xi$
23:      **return** $\gamma$

---

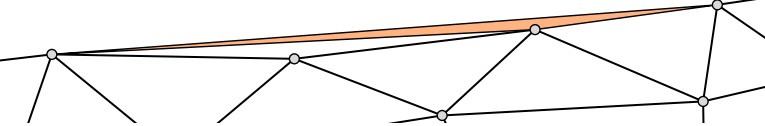

(a) The inside angles of the shaded boundary cell are below the threshold and it is removed.

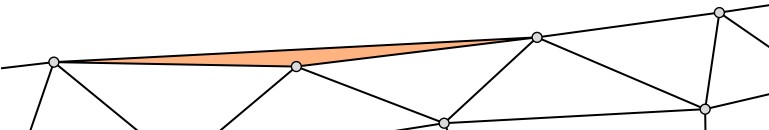

(b) The removal has made this (previously) interior cell into a boundary cell that itself falls below the angle threshold and gets removed as well.

Figure 11: Delaunay triangulation can produce elongated, splinter-like cells when interior nodes are close to the boundary of the convex hull as in this example. We post-process the Delaunay triangulations with Algorithm 1 to filter these cells out.

