# OpenReview forum: "Learning the Dynamics of Physical Systems from Sparse Observations with Finite Element Networks"
_ICLR.cc/2022/Conference — ICLR 2022 Spotlight_

### Official Review · Reviewer_CGtm · 2021-11-01

**Correctness:** 4
**Technical Novelty And Significance:** 3
**Empirical Novelty And Significance:** 4
**Recommendation:** 8
**Confidence:** 4

**Main Review:**

The paper uses Message Passing Neural Networks to implement a Finite Element Method with learnable dynamics. It combines different models and techniques from the literature, but it clearly does that in a nontrivial way including modifications, therefore it is more than just derivative work.

I find it valuable that the author also considers performance of implementation, and discusses consideration on GPU architecture related performance issues. Also the effort put into making the work reproducible adds value to the paper.

The method can clearly have practical value, and the discussion of the method is clear and quite detailed. I am not as familiar with the PDE/NN literature as for example the Neural ODE literature, therefore I cannot rule out entirely that something similar already exists.

Questions / actionable comments:

I)
"The results show that FEN and T-FEN provide the smallest prediction error on both datasets with a further boost due to the separate transport term in T-FEN."
I feel this is a bit too strong statement. While I tend to accept the paper with the results in Table 1 given the other clear benefits of the method (like robustness and interpretability) I am not comfortable with this statement. The table is based on 3 repeats, and have results for example:

Black Sea dataset  CT-MPNN 0.944 ± 0.003  vs.  FEN 0.938 ± 0.005,

these confidence intervals are clearly overlapping and using 3 samples I am a bit skeptical. This is one of the strongest gaps in the table.
In the ScalarFlow FEN is really on par with CT-MPNN. Did you used a statistical test to bold the results?

Again, I have no problem supporting the paper even if the method is on par with the best competitor, but the statement in the paper should be supported by the statistics.

Similarly, in the case of NFE CT-MPNN have approx +-60 deviation on the ScalarFlow dataset, I am not too comfortable to compare these numbers.

II)
What is the motivation of using L1 loss? instead of like MSE?

III) In Experiments/Multi-step Forecasting section:
Time horizon choice is well motivated, but still feels arbitrary in some sense. Does the author see some way to formalize what we accept as "meaningful dynamics"?
How one should choose a comparison horizon for example. Can it be done at least approximately without domain knowledge on the system?

IV) In Model/Network architecture section:
Order invariance of cell vertices is assured by ordering the nodes canonically. Would the author expect improvement if a permutation invariant network, like a Set Transformer be used here? Why or why not?
In some sense the message aggregation step, summation being permutation invariant, is a set network, but not a very expressive one. This  however makes the cell order invariant for a given node and not the other way around.


**Summary Of The Paper:**

The author proposes a method for forecasting in Partial Differential Equations by coupling Finite Element Method on an arbitrary grid with the learning of the dynamics from data. For this purpose a variant of message passing based graph networks is used. It is show in the paper that it's possible to incorporate priors on the structure of the PDE that results in an interpretable solution. The model also show more stability to changes of the mesh structure in test time (like superresolution) and to extrapolation than competitors.


**Summary Of The Review:**

The paper give valuable contribution, the method expected to be practical, robust, and in some cases interpretable. I find the statement on raw prediction error overly strong.

---

> ### Author Response · Authors · 2021-11-23
> **Author Response**
>
> Thank you for the thorough review.
>
> ### Q1) Can you increase the number of samples?
>
> We have increased the number of samples on the Black Sea and ScalarFlow datasets to 10 and recreated the tables and figures. In the updated table, we have bolded the numbers for which the standard deviations "overlap" with the best result.
>
> The NFE and runtimes are now in a separate section in the appendix. Even though the variability of the number of function evaluations is rather high, we still think that they provide valuable information to the reader. They show that the computational complexity of a trained ODE-based model as indicated by the NFE can vary significantly between runs and that, overall, our models and the baseline have comparable runtime costs.
>
> We have adapted our claims according to the additional new results on synthetic data and the variability in the number of function evaluations.
>
> ### Q2) What is the motivation of using L1 loss instead of like MSE?
>
> We noticed early on that training FEN models with an L1 loss works significantly better than mean squared error. The following table compares the MAE achieved over 10 runs of both models trained on Black Sea with L1 and MSE loss respectively.
>
> | Model |           L1 |          MSE |
> |-------|--------------|--------------|
> | FEN   | 0.937+-0.003 | 0.977+-0.015 |
> | T-FEN | 0.936+-0.003 | 0.956+-0.003 |
>
> ### Q3) Is there a principled way to choose the time horizon for multi-step forecasting? Is domain knowledge on the system strictly required?
>
> We are not aware of a way to decide for how many steps a multi-step forecast is possible a-priori. However, time horizons similar to what we chose are common in spatio-temporal forecasting across domains: [de-bezenac18] use 6 steps in sea surface temperature prediction, traffic forecasting is often evaluated over 3, 6, and 12 steps [xu20], and [shi15] forecast precipitation for 15 steps.
>
> [de-bezenac18] Bézenac, Emmanuel de et al. “Deep Learning for Physical Processes: Incorporating Prior Scientific Knowledge.” ArXiv abs/1711.07970 (2018).
> [xu20] Xu, Mingxing et al. “Spatial-Temporal Transformer Networks for Traffic Flow Forecasting.” ArXiv abs/2001.02908 (2020).
> [shi15] Shi, Xingjian et al. “Convolutional LSTM Network: A Machine Learning Approach for Precipitation Nowcasting.” NIPS (2015).
>
> ### Q4) Do you expect an improvement from permutation invariant networks like Set Transformers?
>
> Sorting the nodes to put them in a canonical order achieves a somewhat unstable order invariance. We call it unstable because we concatenate the node features in the sorted order but this order can change due to small changes in orientation of the cell. Imagine an upright triangle with vertices ABC such that the baseline AB is almost horizontal (and the height of C above AB is sufficiently small). If the slope of the baseline is positive, the features are concatenated as ABC but tilting the slope of the baseline just slightly into the negative changes that order to BCA. Therefore, two triangles differing only by a small rotation look very different from the point of view of an MLP. A fully permutation invariant network at this place would avoid this pitfall and we would expect an improvement in performance and generalization to unseen meshes.

---

> > ### Comment · Reviewer_CGtm · 2021-11-28
> > **Response to Authors**
> >
> > Thank you for the detailed answers for my (and all other) reviews. The new experiments and higher sample sizes make the results clear.
> > I update my review accordingly, and I keep my opinion, the paper can be a valuable contribution to the conference.

---

### Official Review · Reviewer_EPhP · 2021-11-01

**Correctness:** 3
**Technical Novelty And Significance:** 3
**Empirical Novelty And Significance:** Not applicable
**Recommendation:** 8
**Confidence:** 4

**Details Of Ethics Concerns:**

None.

**Main Review:**

Strengths:

- The paper is well written and well presented. On the whole it was relatively easy to understand and the diagrams definitely contribute to that.
- The proposed model is well motivated and nicely extends ideas such as Graph Neural ODEs to the PDE domain.
- The paper considers the issues with a naive implementation in depth, (which is that the model would be slow). It then provides the solution to this.
- Rigorous tests on the proposed model are carried out against robust baselines.
- There is an extensive review of related work.
- A lot of effort has been put in to make the results reproducible, including all experimental details and code/datasets to come.

Weaknesses/Questions:

- My main concern is that training has been carried out over 10 time steps, was this a hyperparameter that was tuned? I agree that the correlation over say 30 steps will be minimal, however my understanding is that all models learn an update based on the current state and potentially a few steps in the past. Would it not still be possible to train over more time steps? Could T-FEN extrapolate better than GWN if it is trained on a larger time range?
- One of the proposed reasons GWN outperforms T-FEN at extrapolation is that it can use the past time-steps as input. This could be extended to T-FEN in the form of delay differential equations (https://arxiv.org/abs/2102.10801), where the dynamics takes the state at $(t)$ and the state at $(t-\tau)$ as input. This would make the model use the past as well. Would it be possible to carry out an ablation study using this?
- The paper could benefit from a discussion section. Saying what the model is good at and bad at. For example, we see that it performs very well in the times used to train, but is not as good at extrapolation.
- In table 1, for the ODE based method, the number of function evaluations are provided to show T-FEN is faster than FEN and CT-MPNN. Is it possible to time the evaluation to support this claim further? That way one could also compare to the other two baselines GWN and PA-DGN.
- Is it possible to move the related work section, either to the introduction or to just before the conclusion? It breaks the flow.
- What is the reason for using the $L_1$ loss over the $L_2$ loss? Could we expect better/worse results with $L_2$?
- Could the model be improved even further by using more physically informed terms? For example, like those in electro-magnetism $\dot{v}=qv \times B$, where $B$ is learnt (https://arxiv.org/abs/2109.07359). Or, because the experiments use data of fluids, could it help to include Laplacian or curl terms in the dynamics, which appear in Navier-Stokes? Is it possible to carry out an ablation?
- The mass matrix $A$ is approximately inverted by lumping the matrix, are there situations where this approximation could lead to errors?
- Is it possible to extend this method where we expect higher order PDEs, that include terms such as $\frac{\partial^{2}u}{\partial t\partial x}$?
- The model disentangles the dynamics into a convection term and the remainder. How does this relate to disentangling the dynamics of an Augmented Neural ODE (https://arxiv.org/abs/1904.01681) into a velocity and an acceleration (https://arxiv.org/abs/2006.07220), which can also aid the interpretability of the ODE?
- I don’t entirely understand the Black Sea dataset. Is the mean temperature the mean temperature of the entire sea or of a region, say $1m^2$? If it is of the whole sea, does this not remove the spatial element of the task? Additionally are there any interesting effects appearing over long time periods due to global warming/concept drift? Given that the training regime is taken over 2012-2017 and the testing regime is in 2019?

Minor Points:

- The paragraph just above equation 15 has a small mistake. The manuscript says “This would prohibit training even with the adjoint equation (Chen et al., 2018)”, while the paragraph is about the speed of training. The benefit of the adjoint method is that it is memory efficient but slow compared to directly backpropagating through an ODE solver, which is fast but uses a lot of memory.
- There is a typo at the bottom of page 8, “disappears Extrapolationat the”
- There is a typo at the bottom of page 16, “ODE-base models” should be “ODE-based models”


**Summary Of The Paper:**

This paper proposes a new model for learning partial differential equations from data. The PDE is first discretized then solved as an ODE. The dynamics function is learned with Message-Passing Neural Networks, where the function is split into a sum of physically informed terms. This splitting both improves model performance and makes the model more interpretable by disentangling the dynamics. The model is tested rigorously against multiple baseline models, and the results show the new model performs well.

**Summary Of The Review:**

The paper is well written, the model (to my best knowledge) is novel, with the method building on existing work. The experiments rigorously test the model against the necessary baselines and information is given in the appendices on reproducing the results. Therefore I recommend acceptance, with a few clarifications to be made.

**EDIT** I have increased my confidence score from 3 to 4 after my initial questions have been answered.

---

> ### Author Response · Authors · 2021-11-23
> **Author Response**
>
> Thank you for the thorough review.
>
> ### Q1) How did you arrive at a time horizon of 10? Could we get performance out of training on longer sequences?
>
> The time horizon of 10 steps was an arbitrary decision and we did not optimize over it as a hyperparameter (except for the experiment described in the next paragraph). We decided for 10 steps because it is short enough that not too much uncertainty accumulates from unobserved influences but long enough that the models have to make a prediction over a significant time frame. Time horizons similar to our choice are common in spatio-temporal forecasting across domains: [de-bezenac18] use 6 steps in sea surface temperature prediction, traffic forecasting is often evaluated over 3, 6, and 12 steps [xu20], and [shi15] forecast precipitation for 15 steps.
>
> In an early experiment on ScalarFlow, we saw that increasing the length of the training sequences past 5 did not improve the predictive performance. This makes sense insofar that ODE-based models in the data space like FEN are auto-regressive and, intuitively, cannot do any better than modeling the short-term/instantaneous dynamics as closely as possible. Furthermore, we observed for all models that training them on long sequences stopped learning entirely. Our interpretation is that in real-world spatio-temporal data noise accumulates so much over long time frames that a prediction becomes impossible. Therefore we do not believe that training on longer sequences can squeeze more performance out of FEN.
>
> [de-bezenac18] Bézenac, Emmanuel de et al. “Deep Learning for Physical Processes: Incorporating Prior Scientific Knowledge.” ArXiv abs/1711.07970 (2018).
> [xu20] Xu, Mingxing et al. “Spatial-Temporal Transformer Networks for Traffic Flow Forecasting.” ArXiv abs/2001.02908 (2020).
> [shi15] Shi, Xingjian et al. “Convolutional LSTM Network: A Machine Learning Approach for Precipitation Nowcasting.” NIPS (2015).
>
> ### Q2) Can T-FEN also use past information via delay differential equations?
>
> Combining FEN with delay differential equations would be one way for the model to infer latent, unobserved information such as accelerations from the data. Another would be to include learn dynamics on an embedding space directly as Augmented Neural ODEs [dupont19] or Latent ODEs [rubanova19] do. Unfortunately, torchdiffeq does not support delay differential equations and extending the library in the necessary ways is not possible in the time frame of this discussion period. Nonetheless, the combination of FEN and DDEs is an intriguing suggestion and we will be sure to investigate it in the future.
>
> [rubanova19] Rubanova, Yulia et al. “Latent ODEs for Irregularly-Sampled Time Series.” ArXiv abs/1907.03907 (2019).
> [dupont19] Dupont, Emilien et al. “Augmented Neural ODEs.” NeurIPS (2019).
>
> ### Q3) Can you add a separate discussion section?
>
> Unfortunately, we were unable to fit an extra discussion section into the paper while staying under the page limit.
>
> ### Q4) How do the models compare in terms of wallclock runtime?
>
> GWN provides the fastest evaluation times as it is based on temporal convolutions which are highly optimized and parallelizable. The recurrent PA-DGN is slower by about a factor of three because of its recurrent architecture. ODE-based models are the slowest since they rely on adaptive ODE solvers which most of the time perform much more work per prediction step than other architectures. At the same time, ODE-based models are the only architecture that can naturally model continuous-time data.
>
> ### Q5) The related work section breaks the flow.
>
> We have moved the related work section between the experiments and the conclusion.
>
> ### Q6) What is the reason for using the L1 loss over the L2 loss? Could we expect better/worse results with L2?
>
> We noticed early on that training FEN models with an L1 loss works significantly better than mean squared error. The following table compares the MAE achieved over 10 runs of both models trained on Black Sea with L1 and MSE loss respectively.
>
> |       |           L1 |          MSE |
> |-------|--------------|--------------|
> | FEN   | 0.937+-0.003 | 0.977+-0.015 |
> | T-FEN | 0.936+-0.003 | 0.956+-0.003 |
>
> ### Q7) Can the model be improved further with more physically informed terms? For example $v = qv \times B$ from electromagnetism or the Laplacian and curl terms from Navier-Stokes.
>
> Extending FEN models with more physically informed terms should be possible, though each equation needs to be considered on a case by case basis. It is, for example, easy enough to derive a special form of FEN based on the wave equation. On the other hand, if the equation were to include third-order derivatives, we would need higher-order basis functions which are a topic for future research. Overall, combining FEN with more advanced PDEs is a topic that we want to investigate.

---

> ### Author Response · Authors · 2021-11-23
> **Author Response contd.**
>
> ### Q8) Can mass-lumping lead to bad errors?
>
> From a practical perspective, mass lumping stabilizes model training empirically. When we trained models with the exact mass matrix, learning would often fail. We suspect that the reason lies in small triangulation cells which can occur in real data when you have no control over the sensor locations. Empirically, these small cells increase the condition number of the mass matrix while lumping decreases the condition number. For example, on Black Sea with 1000 nodes the original mass matrix has a condition number of $\operatorname{cond}(A) \approx 46.9$ while the lumped mass matrix has $\operatorname{cond}(\tilde{A}) \approx 29.9$. This difference means that the solution of Equation (11) is more noise-tolerant with the lumped mass matrix which means better gradients.
>
> Theoretically, mass lumping is known to introduce dispersive effects [guermond13]. However, we have not observed errors or training failures which we would attribute to dispersion. We conjecture that in real-world data noise overshadows the adverse effects of mass lumping. On noise-free simulated data however, these effects might become significant and advanced approximations or correction methods would be warranted.
>
> [guermond13] Guermond, Jean-Luc, and Richard Pasquetti. "A correction technique for the dispersive effects of mass lumping for transport problems." Computer Methods in Applied Mechanics and Engineering 253 (2013): 186-198.
>
> ### Q9) Can this method be extended to higher-order PDEs with terms such as $\frac{\partial^2 u}{\partial t \partial x}$?
>
> The model in its current form with piece-wise linear basis functions can be extended with terms of up to second order. Second order terms have to be handled via Gauss' theorem as we have used in the derivation of the transport term in Appendix A. Terms of order three and up would require higher-order basis functions which we want to investigate in future work. Second order basis functions, for example, would additionally increase the smoothness of the prediction but they are non-trivial to implement because of the additional degrees of freedom.
>
> Terms mixing spatial and temporal derivatives are however problematic on a more basic level for FEN, because the method of lines requires that the PDE can be written in the form of Equation (1) with the time-derivative separated from the rest which is not possible if it includes a term like $\frac{\partial^2 u}{\partial t \partial x}$. (Technically, you can separate any variable. In 2D-space you could also semi-discretize in $t$ and $x$ and then forecast along the $y$-axis with $y$ taking the role of time)
>
> ### Q10) How does your disentangling of dynamics compare to the disentangling of dynamics into velocity and acceleration in second-order neural ODEs (a form of augmented neural ODEs)?
>
> These two types of disentanglement are orthogonal and could be combined in future work. FEN splits the dynamics of the observed data into additive parts. Augmented neural ODEs embed the dynamics into a (learned) latent space, which for SONODE takes the role of an acceleration by construction. In principle, it should be possible to have additive parts and latent state in the same model. However, with FEN we want to exploit prior knowledge on the form of the underlying PDE. More research is required to combine this approach with a learned latent space where the features in general do not correspond to physical quantities.
>
> ### Q11) What is the temperature averaged over in the Black Sea dataset? Did you observe climate change or concept drift as the dataset spans multiple years?
>
> The dataset provides daily average values on a grid of size 395x215. The temperature measurement at a grid node represents the daily average temperature of a 1/27° x 1/36° section of the Black Sea. You can see an example of the spatial variability of the daily temperature means sampled at coarse and fine resolution in Figure 8 in the appendix.
>
> We did not observe any trends or concept drift in the data. Over timeframe of just 8 years it is not unlikely that a global long-term trends was overpowered by short-term year to year variations.

---

> ### Comment · Reviewer_EPhP · 2021-11-23
> **Thank you for the answers, I have increased my confidence score.**
>
> Thank you to the authors for their detailed response, not just to my review but all reviews. I am satisfied with the answers provided and as such have changed my confidence score to 4/5.
>
> I would still recommend having even just a paragraph of discussion, either in the appendix, or if the paper is accepted using the additional space for it.

---

> > ### Author Response · Authors · 2021-11-30
> > **Response**
> >
> > Thank you, we will add a discussion section to the appendix for the final version of the paper or in the additional space if it is granted.

---

### Official Review · Reviewer_Zk77 · 2021-11-02

**Correctness:** 4
**Technical Novelty And Significance:** 4
**Empirical Novelty And Significance:** 4
**Recommendation:** 8
**Confidence:** 3

**Main Review:**

# Strenghts

- The paper looks at a very realistic setting for learning PDEs from data: a finite number of samples and (partially) unknown true dynamics. Tackling these problems concomitantly is of great practical importance and this makes the proposed method relevant for practical applications.
- The introduction does a good job motivating the work and pinpointing the main challenges of learning dynamics from data.
- As someone with little experience with the finite element method, Section 2.1 does a great job explaining the required background in the right amount of detail for understanding the paper.
- The connection that the paper makes between the finite element method and message passing neural networks is interesting and, to the best of my knowledge, original.
- The authors show how inductive biases can be added to the model by using a certain prior over the structure of the function $F$.
- I like that the paper focuses on real-world datasets. Also, the datasets themselves are extremely interesting to visualize and make the paper more interesting.
- Figure 3 provides a very insightful qualitative understanding of the proposed model compared to the baseline.
- I am glad that the paper includes a super-resolution experiment. Often, models that work with a discretised space can be very sensitive to changes in the resolution of the mesh. Figure 4 shows that the proposed model is relatively robust to changes in the number of triangles.
- The authors show (in a relatively specific setting) that factorizing the dynamics achieves a disentanglement effect which allows some degree of interpretability of the model.

# Weaknesses

- While I appreciate the focus on real-world datasets, a synthetic experiment where the model could have been evaluated in a more systematic way would have been useful.
- The trick to stabilize training described at the end of Section 3 is slightly peculiar. How important is this trick? Do the authors have any results for when this trick is not used? Could this trick improve the performance of the baselines as well?
- The importance of the approximation from Equation (13) is not studied. Perhaps that is something that could have been tried in the more controllable synthetic setting I was suggesting above. In general, I would be interested to know what are the costs of this approximation and if more advanced approximations might be worth being considered to boost performance.
- Minor suggestion: The paper frames the model as a hypergraph neural network. However, the authors might want to be aware that there is a recent line of work developing simplicial and cell complex neural networks: https://arxiv.org/abs/2103.03212 (ICML 2021), https://arxiv.org/abs/2106.12575 (NeurIPS 2021), https://arxiv.org/abs/2010.03633. Since the model learns a function over the 2-simplices in the simplicial complex, the model is probably more accurately described as a type of simplicial neural network.

**Summary Of The Paper:**

The paper proposes a graph / simplicial neural network based on the finite element method for learning dynamics from data when only a finite number of samples exist and the true dynamics are not known or only partially known.

**Summary Of The Review:**

The weaknesses reported above are relatively minor and far outweighed by the strengths of the paper. Therefore, I recommend the paper for acceptance.

---

> ### Author Response · Authors · 2021-11-23
> **Author Response**
>
> Thank you for the thorough review.
>
> ### Q1) Can you evaluate FEN on synthetic data?
>
> We have extended the multi-step forecasting section with an evaluation on the CylinderFlow dataset from [pfaff21]. This dataset consists of a collection of simulated flows along a 2D channel around a cylinder shaped obstacle. It is interesting because the flow can be stable or exhibit periodic behavior depending on the in-flow conditions and the size and position of the cylinder.
>
> Due to the size of the dataset, we were only able to finish three training runs of each model before the end of the discussion period, but we will increase the number of samples to 10 in a final version of the paper. The main findigs are that 1) FEN achieves the best performance among the five models on noise-free simulated data, 2) since the dataset includes no attribute that is transported (only velocity and pressure), T-FEN's assumption is violated and the performance worse, 3) CT-MPNN training on this dataset has very high variance. While FEN and T-FEN converge to more or less the same performance every time, CT-MPNN mostly gets stuck in local minima. One run managed to learn meaningful dynamics but could not be evaluated, because the learned dynamics were too expensive in terms of number of function evaluations and therefore runtime to finish in time.
>
> A thorough investigation of the efficacy of FEN on synthetic data generated from various known dynamics will be the topic of future research work.
>
> [pfaff21] Pfaff, Tobias et al. “Learning Mesh-Based Simulation with Graph Networks.” ICLR (2021).
>
> ### Q2) How important is it to start training with short sequences and then continually increase the length? Do the authors have any results for when this trick is not used? Could this trick improve the performance of the baselines as well?
>
> In our experience, this technique stabilizes the training but does not improve the final performance of the model on real-world data. Training on sequences of the target length from the beginning sometimes can get models stuck in local minima for varying numbers of epochs whereas with this technique training progress from epoch to epoch is more consistent. The developers of DiffEqFlux.jl also recommend this technique to avoid local minima in Neural ODE training and give a synthetic example with periodic behavior which is difficult to learn without this trick [diffeqflux-1]. This technique can also be seen as a variant of multiple shooting in data space [diffeqflux-2,turan21].
>
> We have used this with the CT-MPNN baseline as well in the reported results to improve its training stability, in addition to the other small improvements to CT-MPNN's configuration described in the appendix. Neither PA-DGN nor GWN would get stuck in local minima during training, so we did not apply it to these two baselines.
>
> [diffeqflux-1] https://diffeqflux.sciml.ai/dev/examples/local_minima/
> [diffeqflux-2] https://diffeqflux.sciml.ai/dev/examples/multiple_shooting/
> [turan21] Turan, Evren M. and Johannes Jäschke. “Multiple shooting with neural differential equations.” ArXiv abs/2109.06786 (2021).

---

> > ### Comment · Reviewer_Zk77 · 2021-11-27
> > **Response to Authors**
> >
> > Thank you for your detailed response and additional synthetic experiments! Please include a short version of these clarifications in the final version of the paper.
> >
> > Overall, I remain of the opinion that this would be a good contribution to the conference and I will maintain my acceptance score.

---

> > > ### Author Response · Authors · 2021-11-30
> > > **Response**
> > >
> > > Thank you, we will include the gist of our clarifications in the appendix of the final version of the paper.

---

> ### Author Response · Authors · 2021-11-23
> **Author Response contd.**
>
> ### Q3) What is the importance of mass lumping in Equation (13)?
>
> From a practical perspective, mass lumping stabilizes model training empirically. When we trained models with the exact mass matrix, learning would often fail. We suspect that the reason lies in small triangulation cells which can occur in real data when you have no control over the sensor locations. Empirically, these small cells increase the condition number of the mass matrix while lumping decreases the condition number. For example, on Black Sea with 1000 nodes the original mass matrix has a condition number of $\operatorname{cond}(A) \approx 46.9$ while the lumped mass matrix has $\operatorname{cond}(\tilde{A}) \approx 29.9$. This difference means that the solution of Equation (11) is more noise-tolerant with the lumped mass matrix which means better gradients.
>
> Theoretically, mass lumping is known to introduce dispersive effects [guermond13]. However, we have not observed errors or training failures which we would attribute to dispersion. We conjecture that in real-world data noise overshadows the adverse effects of mass lumping. On noise-free simulated data however, these effects might become significant and advanced approximations or correction methods would be warranted.
>
> [guermond13] Guermond, Jean-Luc, and Richard Pasquetti. "A correction technique for the dispersive effects of mass lumping for transport problems." Computer Methods in Applied Mechanics and Engineering 253 (2013): 186-198.
>
> ### Q4) Is your model actually a simplical neural network?
>
> Thank you for bringing these recent works on simplical neural networks to our attention. As far as we understand, a simplical complex is closed under taking subsets which would mean that a mesh hypergraph as a simplical complex would also include pairwise edges as well as the individual nodes as edges. However, FEN only works with mesh cells, but not with edges or nodes because these have 0 volume and therefore do not influence the dynamics. For this reason we are uncomfortable introducing FEN as a simplical neural network as of now and decided to stay with the framing as a hypergraph neural network, which might be overly general but is definitely correct.

---

### Official Review · Reviewer_4K4v · 2021-11-02

**Correctness:** 3
**Technical Novelty And Significance:** 4
**Empirical Novelty And Significance:** 2
**Recommendation:** 6
**Confidence:** 4

**Main Review:**

This manuscript proposes a new graph neural net (GNN) method to learn the dynamics of a spatio-temporal PDE-driven dynamical system directly from data. The authors propose to do that using the finite element method (FEM). The proposed method builds on using: basis function approximation for the (unknown) field u, Galerkin method with the assumption that discrepancy between the dynamics F and basis function approximation is orthogonal to (finite) basis functions, method of lines, and message passing GNN as a proxy for the dynamics. The use of linear interpolation allows to express the time derivative of $Y$ as a solution of a system of linear equations, which is further approximated to gain computational additional efficiency. Authors also propose a method to incorporate inductive bias into model learning for models that are assumed to contain a convection component. Overall the proposed method is well-motivated, and for the most part the description is clear. To my knowledge, the proposed method is novel and contains some methodologically new ideas, and the performance seems to be on par with previous methods that learn free-form dynamics, and shows an improvement for models that contain a convection component when such prior knowledge is utilised in the model training. Authors could address and/or clarify the following aspects:
1. I understand that a piece-wise linear basis simplifies computational complexity by making some of the computational steps straightforward, but on the other hand the selected basis is the simplest and obviously not optimal from approximation accuracy point of view. Can the method be extended to other basis? For example, if we knew that the dynamics $F$ contains a diffusion term $\nabla^2u$ we would not be able to introduce it since the second derivative of PWL functions is zero everywhere. I think the discussion of possible limitations of the PWL basis and of possible extensions to higher-order bases is missing.
2. It seems that the measurement from the initial time point is used as the initial state? Why not introduce a separate free parameter for the initial state?
3. As described towards the end of the manuscript, the system is initialed with an initial state and then the PDE dynamics define how the system evolves over time. However, below eq. (12) it is noted that "where $u$ encodes the data $Y$ at time $t$ in its coefficients." Is $u$ defined based on the dynamically evolving system state or by data?
4. Paragraph "Network architecture" on page: scimitar-learn is used to compute inner product between basis functions. Provide a brief description of how the computation is done.
5. End of page 6: gradients are back propagated through an ODE solver. Why not use adjoint method (possibly with checkpoints) as proposed in previous work and implemented e.g. in comparison methods? Discrete back-propagation may not scale to longer sequences. Is this the reason why data trajectories are shortened?
6. Table 1: split the methods into two groups. Group 1 should includes PA-DGN, GWN, CT-MPNN and FEN, which do not assume prior knowledge about system dynamics. FEN performs similarly with CT-MPNN on ScalarFlow and perhaps slightly better on Black Sea data (MAE values within one std). Group 2 should include only T-FEN that is specifically designed to learn convection systems, and thus provides a small performance improvement.
7) In experiments, the authors make their models time- and position-dependent while the strongest baseline models (CT-MPNN) does not utilize neither time nor positions. That makes it hard to tell whether the improvements in performance of FEN and T-FEN are due to the models's structure and inductive biases or due to time and position dependencies. Authors should provide an ablation study to address this.
8. Provide additional comparisons on systems with larger variety of dynamics using simulated data (for which the ground truth is known) to better understand when FEN performs better/worse than comparison methods.
9. Datasets are subsampled to 1000 spatial points. Provide results for smaller and also larger spatial grids to demonstrate the argument that "approximation becomes arbitrarily good as the mesh resolution increases".

**Summary Of The Paper:**

This manuscript proposes a new graph neural net (GNN) method to learn the dynamics of a spatio-temporal PDE-driven dynamical system directly from data.

**Summary Of The Review:**

To my knowledge, the proposed method is novel and contains some methodologically new ideas, and the performance seems to be on par with previous methods that learn free-form dynamics, and shows an improvement for models that contain a convection component when such prior knowledge is utilised in the model training.

---

> ### Author Response · Authors · 2021-11-23
> **Author Response**
>
> Thank you for the thorough review.
>
> ### Q1) Can FEN be extended to higher-order basis functions?
>
> We believe that an extension to higher-order basis functions is possible, though not trivial, and an aspect that we want to explore in future work. From our perspective, the main problem is that higher-order basis functions require more support points as the degrees of freedom of the basis functions increases. Second order polynomials in two dimensions, for example, require 6 support points within each cell. This is no issue in simulations where we can choose evaluation points, but becomes a problem when modeling real sensor data where the positions of the sensors are already fixed. With piece-wise linear basis functions, any triangular mesh of the sensor locations such as a Delaunay triangulation will by definition have the three required nodes in each cell. If we want second-order bases, we have to triangulate the sensors in such a way that each cell contains exactly 6 sensors which is itself a non-trivial problem.
>
> The reward for this challenge would be smooth predictions and the ability to incorporate higher-order derivative terms in the unknown dynamics $\mathcal{F}$. Second order terms like $\nabla^2 u$ can, however, already be included with linear bases. The trick is to "move" one derivative from the trial function to the test function similar to the derivation in Appendix A. Then we are left with just inner products of first derivatives and get non-zero weights even with linear bases.
>
> ### Q2) Why is the initial measurement used as the initial model state? Couldn't we have the state be a latent variable?
>
> With FEN we decided to model the dynamics of the data in data space. In principle, FEN could just as well model dynamics in a latent space in a kind of encoder-decoder architecture similar to [latent-ode]. However, the measurements usually correspond to physical quantities where we can reasonably argue whether their dynamics can be described by a PDE and what form that might take. Introducing a latent space also introduces a level of indirection that makes such reasoning about the dynamics and therefore learning with FENs more difficult.
>
> At the same time, we believe that the inclusion of latent information into FEN is a worthwhile direction for future research. Modeling the data dynamics directly as FEN does means that information can only be carried through time via measured system properties. Additional latent state could enable the model to estimate and work with accelerations, for example, even though only velocities have been measured.
>
> [latent-ode] Rubanova, Yulia et al. “Latent Ordinary Differential Equations for Irregularly-Sampled Time Series.” NeurIPS (2019).
>
> ### Q3) Is $u$ defined based on the dynamically evolving system state or by data?
>
> The coefficients of $u$ encode the dynamically evolving system state, except for $u_0$ where we seed the system state with the data $y^{(0)}$ that the model is conditioned on. We have replaced "data" in the sentence following Equation (12) with "predicted features" to clarify this distinction.
>
> ### Q4) How does scikit-fem compute the inner product between basis functions?
>
> Because the P1 basis functions are piece-wise first-order polynomials, the inner products are integrals over at most second-order polynomials over triangular cells. These integrals can be evaluated exactly numerically, usually via a mapping onto a reference element (the 2-simplex in our case) and the Gaussian quadrature rule. While this procedure does not require too many lines of code to implement, keeping track of the mappings and change of variable is cumbersome, so we preferred to use scikit-fem for this.

---

> ### Author Response · Authors · 2021-11-23
> **Author Response Pt. 2**
>
> ### Q5) Why are you not using the adjoint method for backpropagation, possibly with checkpointing? Is this the reason why you only train on short sequences?
>
> As we argued in the paper, long term predictions on spatio-temporal data from real-world systems are difficult, because they are influenced by many unobserved / unmeasured effects. These unobserved influences accumulate and lead to an increasingly non-deterministic relationship between the input data and the prediction target as the distance in time increases. This is especially apparent on the Black Sea dataset. Yet, our models as well as the baselines fundamentally assume a deterministic setting. The induced non-determinism either prevented the models from learning anything or made them learn just a diffusion process without any interesting or relevant predictions when trained on long sequences. We interpret this to mean that learning these long-term predictions is so difficult or even impossible, that "the only winning move is not to play" for the models.
>
> A feasible alternative is accurate modeling of the short-term dynamics where unobserved effects have not accumulated enough influence on the data yet to make a prediction impossible. We opted for 10-steps of forecasting which is in line with other works in this field [de-bezenac]. As a consequence we did not need the constant-memory requirements of the adjoint method and also avoided the runtime costs that come with it.
>
> [de-bezenac] Bézenac, Emmanuel de et al. “Deep Learning for Physical Processes: Incorporating Prior Scientific Knowledge.” ArXiv abs/1711.07970 (2018): n. pag.
>
> ### Q6) The methods in Table 1 should be grouped by the assumptions they make on the dynamics.
>
> Grouping the methods by assumptions on the dynamics is not so clear cut, because PA-DGN has an extension for diffusion and one could argue that the application of FEN itself already makes the assumption that the dynamics can be formulated in the form of a PDE. Therefore we decided to omit the dividing line entirely from the table.
>
> ### Q7) Are the improvements just coming from extra time- and position information available to FEN and T-FEN?
>
> We have provided time information on Black Sea to all models as an extra feature because there the dynamics clearly depend on the time of the year. On ScalarFlow on the other hand no model has time information because there the actual time is not meaningful. We did not provide the absolute node positions to the CT-MPNN baseline because Iakovlev et al. only used relative positional information between nodes in their paper.
>
> With FEN the inclusion of time- and/or position information is explicitly a modeling decision. On the synthetic CylinderFlow dataset we have decided not to provide position information to any model because the dynamics are fundamentally stationary. In this case both FEN and T-FEN beat the CT-MPNN baseline convincingly.
>
> ### Q8) Can you evaluate FEN on synthetic data?
>
> We have extended the multi-step forecasting section with an evaluation on the CylinderFlow dataset from [pfaff21]. This dataset consists of a collection of simulated flows along a 2D channel around a cylinder shaped obstacle. It is interesting because the flow can be stable or exhibit periodic behavior depending on the in-flow conditions and the size and position of the cylinder.
>
> Due to the size of the dataset, we were only able to finish three training runs of each model before the end of the discussion period, but we will increase the number of samples to 10 in a final version of the paper. The main findigs are that 1) FEN achieves the best performance among the five models on noise-free simulated data, 2) since the dataset includes no attribute that is transported (only velocity and pressure), T-FEN's assumption is violated and the performance worse, 3) CT-MPNN training on this dataset has very high variance. While FEN and T-FEN converge to more or less the same performance every time, CT-MPNN mostly gets stuck in local minima. One run managed to learn meaningful dynamics but could not be evaluated, because the learned dynamics were too expensive in terms of number of function evaluations and therefore runtime to finish in time.
>
> A thorough investigation of the efficacy of FEN on synthetic data generated from various known dynamics will be the topic of future research work.
>
> [pfaff21] Pfaff, Tobias et al. “Learning Mesh-Based Simulation with Graph Networks.” ICLR (2021).

---

> ### Author Response · Authors · 2021-11-23
> **Author Response Pt. 3**
>
> ### Q9) How do FEN and T-FEN perform on finer and coarser meshes?
>
> Our super-resolution experiments in Figure 4 and the appendix show how FEN and T-FEN generalize to finer meshes. We observe a good transfer to meshes with up to 5000 nodes on both datasets. In particular, the rate of deterioration is markedly lower than for CT-MPNN.
>
> Our claim that "the approximation becomes arbitrarily good as the mesh resolution increases" refers to approximating any smooth function with a step function as used in Riemann integrals for example and is not specific to our model.

---

### Decision · Program_Chairs · 2022-01-20

**Decision:**

Accept (Spotlight)

**Comment:**

This paper introduces a graph neural network (GNN) based on the finite element method (FEM) for learning partial differential equations from data. The proposed finite element network is based on a piecewise linear function approximation and a message passing GNN for dynamics' prediction. The authors also propose a method to incorporate inductive bias when learning the dynamical model, e.g. including a convection component.

The paper received three clear accept and one weak accept recommendations. The reviewers discussed the possible extensions of the method, and also raise several concerns regarding experiments, e.g. the added value of a synthetic dataset, implementation tricks or hyper-parameter settings. The rebuttal did a good job in answering reviewers' concerns: after rebuttal, there was a consensus among reviewers to accept the paper.

The AC's own readings confirmed the reviewers' recommendations. The paper is well written and introduces solid contribution at the frontier of GNNs and finite elements methods, especially a pioneer graph-based model for spatio-temporal forecasting derived from FEM. Therefore, the AC recommends acceptance.